# Genetic and epigenetic features direct differential efficiency of Xist-mediated silencing at X-chromosomal and autosomal locations

Agnese Loda [1,7], Johannes H. Brandsma[2], Ivaylo Vassilev[3], Nicolas Servant[3], Friedemann Loos[4], Azadeh Amirnasr[1], Erik Splinter[5], Emmanuel Barillot [3], Raymond A. Poot[2], Edith Heard[6] & Joost Gribnau[1]

Xist is indispensable for X chromosome inactivation. However, how Xist RNA directs chromosome-wide silencing and why some regions are more efficiently silenced than others remains unknown. Here, we explore the function of Xist by inducing ectopic Xist expression from multiple different X-linked and autosomal loci in mouse aneuploid and female diploid embryonic stem cells in which Xist-mediated silencing does not lead to lethal functional monosomy. We show that ectopic Xist expression faithfully recapitulates endogenous X chromosome inactivation from any location on the X chromosome, whereas long-range silencing of autosomal genes is less efficient. Long interspersed elements facilitate inactivation of genes located far away from the Xist transcription locus, and genes escaping X chromosome inactivation show enrichment of CTCF on X chromosomal but not autosomal loci. Our findings highlight important genomic and epigenetic features acquired during sex chromosome evolution to facilitate an efficient X chromosome inactivation process.

[1] Department of Developmental Biology, Erasmus University Medical Center, Wytemaweg 80, 3015 CN Rotterdam, The Netherlands. [2] Department of Cell Biology, Erasmus University Medical Center, Wytemaweg 80, 3015 CN Rotterdam, The Netherlands. [3] Bioinformatics and Computational Systems Biology of Cancer, INSERM U900, Paris 75005, France. [4] Equipe 11 labellisée par la Ligue Nationale contre le Cancer, Centre de Recherche des Cordeliers, Paris 75006, France. [5] Cergentis B.V., Padualaan 8, 3584 CH Utrecht, The Netherlands. [6] Mammalian Developmental Epigenetics group, Institut Curie, CNRS UMR 3215, INSERM, U934 Paris, France. [7] Present address: Mammalian Developmental Epigenetics group, Institut Curie, CNRS UMR 3215, INSERM, U934 Paris, France. Correspondence and requests for materials should be addressed to J.G. (email: j.gribnau@erasmusmc.nl)

In mammals, dosage compensation of sex chromosomal genes between females (XX) and males (XY) is achieved through X chromosome inactivation (XCI). XCI starts with the mono-allelic upregualtion of the X-linked non-coding gene *Xist* and culminates in the conversion of one entire X chromosome into a silent heterochromatic entity known as the Barr body (Xi)[1, 2]. During XCI, Xist RNA spreads in *cis* and recruits a multitude of factors involved in trancriptional inactivation. The Xi is initially depleted of euchromatic histone modifications such as H3K4me2/me3 and H3/H4 acetylation, and subsequently enriched for repressive marks such as H3K27me3 and H2AK119ub[3–6]. Although *Xist* is the major player of the process[7–10], the molecular mechanisms by which Xist RNA spreads along the chromosome and triggers gene silencing remains largely unknown. Several X;autosome translocation studies showed incomplete inactivation of the autosomal material[11–14], suggesting a sequence-specific model for Xist spreading. In this context, long interspersed elements (LINE) have been proposed to work as "way stations" for X-linked-specific Xist spreading[15]. LINEs are enriched on both human and mouse X chromosomes relative to autosomes and take part in the formation of the Xi silent compartment[3, 16]. However, LINE-rich genomic areas correspond to gene-poor areas, whereas gene-rich areas are depleted of LINEs. Thus, the preferential spreading of Xist RNA in LINE-rich regions of autosomes may reflect negative selection against cells in which Xist-mediated silencing leads to functional aneuploidy[14, 16–18]. In addition, specific X-linked regions that are targeted by Xist RNA at the onset of XCI are not enriched for LINEs, thus questioning their role in conferring X chromosome specificity to Xist spreading[19, 20].

Around 3–7% of mouse X-linked genes escape from XCI[21]. To date, how these genes can remain transcriptionally active within the silent Xi remains largely unknown. Escaping genes lack the epigenetic marks typical of inactivated genes and retain active marks[22, 23]. Furthermore, they are located outside the Xist RNA domain[3, 24], have been suggested to be intrinsically competent to resist XCI[25], and to be flanked by *cis*-acting elements that protect neighboring genes from escape[26]. CTCF has been proposed to play a role in XCI escape both acting as a boundary element between active and inactive loci[27] or as an anchor that allows looping out of active domains from the Xi territory[28, 29].

Unraveling Xist's functions is critical for a complete understanding of XCI. Here, to address the mechanism(s) directing Xist-mediated silencing, we set up a doxycycline-responsive *Xist* expression system in mouse embryonic stem cell (ESC) lines. By inducing ectopic XCI from several genomic regions in karyotypically normal and abnormal ESC lines, we discovered that: (I) Xist's silencing efficiency is locus dependent, (II) specific X-linked but not autosomal loci are intrinsically prone to become inactivated or to escape XCI, (III) LINEs facilitate gene silencing but are unlikely to work as X-specific way stations, and (IV) CTCF plays a X-specific role in directing XCI escape.

## Results

**An inducible *Xist* expression system in mouse ESC lines**. To assess the efficiency of Xist-mediated silencing from several genomic contexts, we set up a doxycycline-responsive expression system in F1 2–1 ESC lines (129/Sv-Cast/Ei). First, we generated an Xist-inducible transgene using a Cast/Ei bacterial artificial chromosome (BAC) covering 300 kb of the X chromosome including the *Xist* endogenous locus. Through homologous recombination in bacteria[30], 1 kb upstream of *Xist* transcription starting site (TSS) was replaced with a targeting cassette carrying a $P_{tight}$ bidirectional doxycycline-responsive promoter driving Xist and a DsRed reporter gene (Fig. 1). Next, the Xist-inducible

transgene was transfected into F1 2–1 female ESCs in which the reverse tetracycline transactivator M2rtTA had been targeted at the *ROSA26* locus (Supplementary Fig. 1A, B). Neomycin-resistant ESCs were screened by either restriction fragment length polymorphism (RFLP)-PCR or DNA fluorescent in situ hybridization (FISH), and four sets of transgenic female ESC lines were selected: (1) Tg-E clones, in which the Xist transgene was targeted to the *Xist* endogenous locus of the Cast/Ei X chromosome of a wild-type ESC line (40,XX). Four independent Tg-E clones were generated (68, 87, 77, 64), in which successful homologous recombination of the transgene was assessed by RFLP-PCR (Fig. 1a; Table 1). (2) Tg-X;8 and (3) Tg-X clones, in which the Xist transgene was randomly integrated on either X chromosomes of a karyotypically abnormal ESC line carrying two intact copies of chromosome 8 and a duplication of the distal two-thirds of the Cast/Ei chromosome 8, which is fused to the 129/Sv X chromosome in the resulting X;8 translocation (40,XX,t(X;8)) (Supplementary Fig. 1C). We generated three independent Tg-X;8 clones (339, 203b, 267), with the Xist transgene integrated at different loci on the X;8 translocation product, and four independent Tg-X clones (85, 86, 109, 190) carrying the Xist transgene on the wild-type Cast/Ei X chromosome (Fig. 1d, e; Table 1). Finally (4), we generated six independent Tg-12 clones (251, 292, 228, 160, 55, 273), in which the Xist transgene was randomly integrated at different loci of one copy of chromosome 12 in a trisomic ESC line (41,XX,dup12) (Fig. 1f, g; Table 1). All Tg-X;8, Tg-X, and Tg-12 clones were initially selected by DNA FISH using chromosomes X, 8, and 12-specific probes, followed by targeted locus amplification[31] to determine the exact site of integration (Table 1).

**Ectopic Xist RNA spreads in *cis* on chromosomes**. Next, we asked whether ectopic *Xist* could be efficiently expressed upon doxycycline induction and whether the induced Xist RNA could spread in *cis* on chromosomes X, 12, and 8. To this end, all ESC clones were grown in ESC medium supplemented with doxycycline for 5 days. By inducing ectopic XCI in undifferentiated ESCs, we were able to uncouple Xist function from cell differentiation, thus allowing the efficiency of Xist spreading to be assessed independently of any selection on cell viability. Xist RNA could be ectopically expressed from both the X chromosome and autosomes, with the inducible system showing no leakiness of *Xist* expression (Fig. 2a–d). Tg-X;8 clone 267 and Tg-12 clone 160 carry a previously described *Tsix-Stop* allele[32] on the 129/Sv X chromosome, which explains the higher level of basal *Xist* expression in untreated conditions. This Tsix-Stop allele is present in the original lines as a floxed allele and was removed in all clones except clones 267 and 160 upon Cre-mediated removal of the neo selection cassette. Xist RNA enrichment in doxycycline-treated cells vs. untreated cells varies from 10- to 250-fold in between different clones (Fig. 2a–d). In spite of this variability, the enrichment of ectopic Xist in ESCs is either comparable or higher than the one reached by endogenous *Xist* upon neuronal differentiation of untreated ESCs (Supplementary Fig. 2A). In fact, endogenous Xist is upregulated by 3- to 70-fold between day 2 and day 4 of differentiation, when XCI starts, compared to undifferentiated ESCs, prior to XCI (Supplementary Fig. 2A). To assess whether ectopic Xist RNA is stable and can spread in *cis* from different loci, we performed Xist RNA FISH on all transgenic clones after 5 days of doxycycline induction (Fig. 2e–k). About 50–70% of induced cells showed an Xist-coated chromosome (Fig. 2e–h) which was stable over time (Supplementary Fig. 2B, C). Taken together, this data demonstrate that we set up a robust *Xist* expression system to separate Xist RNA function from genomic context and cell differentiation and without any aneuploidy-related phenotype.

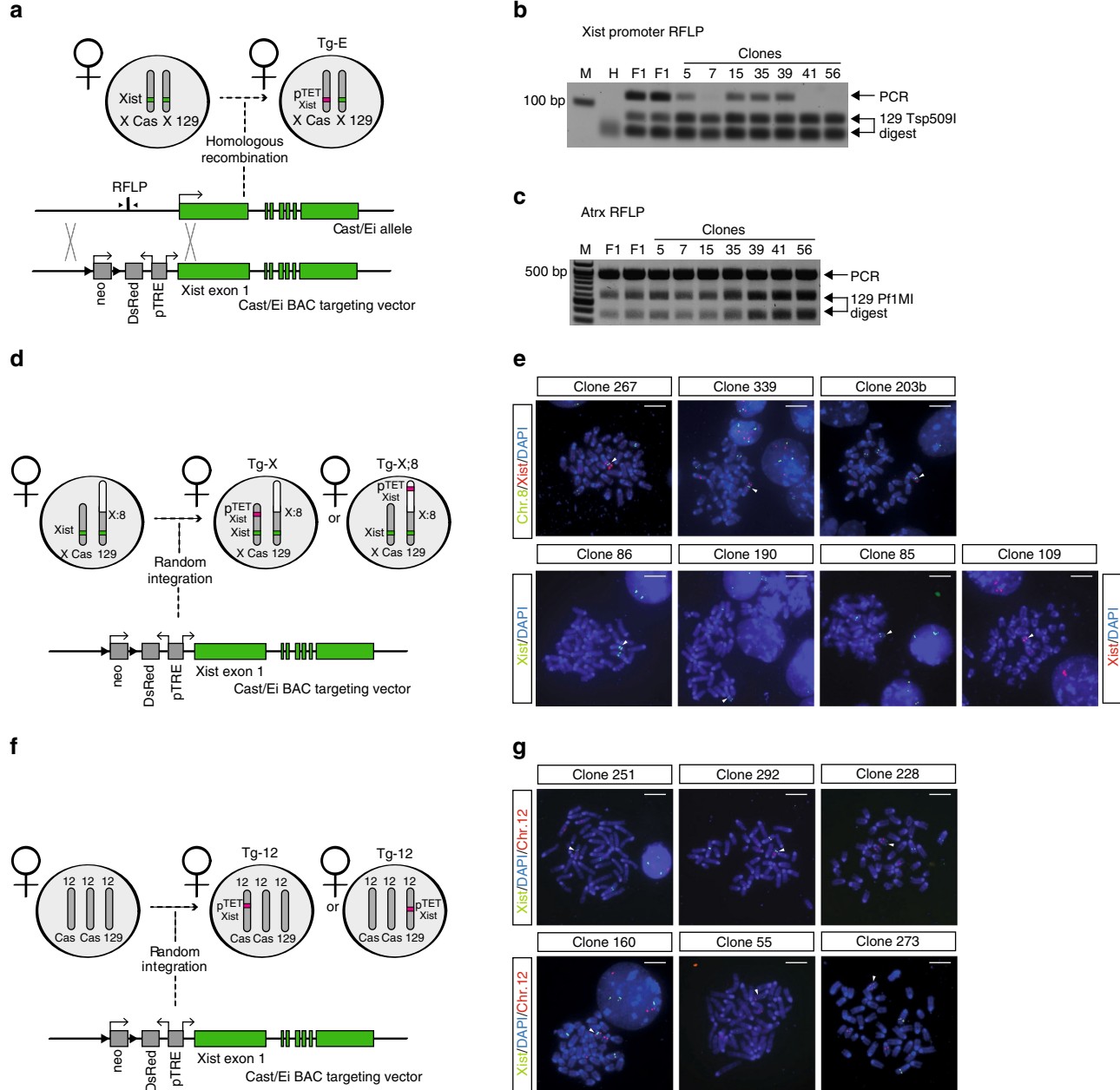

**Fig. 1** Generation of a tetracycline-responsive Xist expression system in ES cells. **a** Targeting strategy to generate Tg-E clones. The Xist TSS was replaced with a bidirectional tetracycline-responsive Ptight promoter, a DsRed reporter gene, and a neomycin resistance cassette. **b** PCR amplification with primers indicated in **a** followed by a Tsp509I RFLP digest of PCR product to identify clones with a correctly targeted Cast/Ei allele. Correct targeting results in loss of Cast/Ei-specific band, as shown for clones 7, 41, and 56. *Arrows* indicate size of PCR product and Tsp509I restriction fragments. *F1* F1 2-1 polymorphic 129/Sv-Cast/Ei mother cell line, M marker, H water. **c** PCR amplification of a fragment of the X-linked gene *Atrx* and *Pf1*MI digest of the PCR product to verify the presence of two X chromosomes. *Arrows* indicate size of PCR product and size of *Pf1*MI restriction fragments. **d, f** Schematic representation of the strategy used to generate Tg-X, Tg-X;8 **d**, and Tg-12 clones **f**. The Xist transgene was randomly integrated into F1 2-1 40,XX,t(X;8) or 41,XX,dup12 ESC lines and neomycin-resistant clones were screened by DNA FISH. **e, g** DNA FISH screening of Tg-X, Tg-X;8 **e**, and Tg-12 **g** clones. *Arrows* indicate the chromosome carrying the Xist transgene. Positive Tg-X;8 clones show co-localization of three DNA FISH signals on the translocated X;8 chromosome corresponding to the Xist endogenous locus, the ectopic transgene, and the chromosome 8 portion of the X;8 translocation. Three independent ES clones were generated (267, 339, 203b). Positive Tg-X clones show two DNA FISH signals on the wild-type Cast/Ei X corresponding to the endogenous Xist locus and to the ectopic transgene, respectively. Four independent ES clones were generated (86, 190, 85, 109). Positive Tg-12 clones show co-localization of two DNA FISH signals on one of the three copies of chromosome 12, corresponding to the Xist-inducible transgene and to chromosome 12, respectively. Six independent ES clones have been generated (251, 160, 273, 292, 55, 228). *Scale bars* represent 5 μm

**X-linked genes are more efficiently silenced than autosomes.** To assess whether ectopic Xist RNA could trigger gene silencing independently of its genomic position, we performed RNA-seq analysis of Tg-E, Tg-X, Tg-X;8 and Tg-12 clones after 5 days of doxycycline treatment in undifferentiated ESCs. We generated

52 RNA-seq libraries including two biological replicates per clone in both doxycycline- treated and untreated conditions. Total RNA-seq reads were aligned to both the 129/Sv and the Cast/Ei parental genomes and the abundance of allele-specific reads was estimated by summing up the number of 129- and Cast-specific

**Table 1 Summary of transgenic clones generated in this study**

| Clone | Category | Chr. | Allele | Karyotype | 5′-3′ Integration sites (mm10) | Additional information |
|---|---|---|---|---|---|---|
| 87 | Tg-E | X | Cast | 40,XX | chrX:103,460,373-103,483,217 | *Xist* promoter 1 kb del (Cast) |
| 68 | Tg-E | X | Cast | 40,XX | chrX:103,460,373-103,483,217 | *Xist* promoter 1 kb del (Cast) |
| 64 | Tg-E | X | Cast | 40,XX | chrX:103,460,373-103,483,217 | *Xist* promoter 1 kb del (Cast) |
| 77 | Tg-E | X | Cast | 40,XX | chrX:103,460,373-103,483,217 | *Xist* promoter 1 kb del (Cast) |
| 267 | Tg-X:8 | X:8 | 129 | 40,XX,t(X;8) | chr8:110,645,220-110,329,507 | Tsix-Stop allele (129) |
| 339 | Tg-X:8 | X:8 | 129 | 40,XX,t(X;8) | ND | – |
| 203b | Tg-X:8 | X:8 | 129 | 40,XX,t(X;8) | chr8:122,340,941-122,343,259 | – |
| 86 | Tg-X | X | Cast | 40,XX,t(X;8) | chrX:134,402,074[a]-134,559,764 | – |
| 190 | Tg-X | X | Cast | 40,XX,t(X;8) | chrX:169,976,922[a]-170,005,736[a] | – |
| 85 | Tg-X | X | Cast | 40,XX,t(X;8) | ~chrX telomere | – |
| 109 | Tg-X | X | Cast | 40,XX,t(X;8) | chrX:103,483,223[a]-103,484,258[a] | – |
| 251 | Tg-12 | 12 | ND | 41,XX,dup12 | ND | – |
| 292 | Tg-12 | 12 | Cast | 41,XX,dup12 | chr12:89,465,956[a]-89,155,600 | – |
| 228 | Tg-12 | 12 | Cast | 41,XX,dup12 | chr12:109,077,348-109,113,528 | – |
| 160 | Tg-12 | 12 | Cast | 41,XX,dup12 | ~chr12 centromere | Tsix-Stop allele (129) |
| 55 | Tg-12 | 12 | 129 | 41,XX,dup12 | chr12:104,543,748[a]-104,556,010 | – |
| 273 | Tg-12 | 12 | Cast | 41,XX,dup12 | chr12:98,483,300[a]-98,489,700 | – |

TLA targeted locus amplification
[a]Due to the features of the integration locus and/or the Xist transgene sequence, the initial TLA experiment did not yield the exact integration site, but did allow to make the approximation indicated

reads of all exonic single-nucleotide polymorphism (SNPs) within the gene[33]. For each gene in our data set, we used the total counts of 129/Sv ($N_{129}$) and Cast/Ei ($N_{Cast}$) allele-specific reads to obtain the ratio of Cast-specific gene expression (Cast/all ratio = $(N_{Cast})/(N_{Cast} + N_{129})$). Only polymorphic sites showing a coverage equal to or higher than five reads were treated as informative. For genes containing a single polymorphic site, the coverage threshold was increased to eight reads. In Tg-E clones, in which the inducible transgene is targeted at the *Xist* endogenous locus of the Cast/Ei X chromosome, the overall X-linked gene expression changes from biallelic expression (ratio = 0.5) in untreated cells to a more 129/Sv-monoallelic gene expression (ratio < 0.5) in doxycycline-treated cells. Since ectopic XCI can be induced in only 60–70% of doxycycline- treated ESCs (Fig. 2e–h), we consider all genes that show a ∆ ratio between 0.1 and 0.5 to be affected by Xist RNA (Fig. 3a, b), resulting in silencing of 60–70% of all X-linked genes (Fig. 3a, b), and highlighting efficient ectopic XCI when Xist is induced from its endogenous locus. Similarly, in Tg-X and Tg-X;8 clones, X-linked gene expression upon doxycycline induction shifts from biallellic (ratio = 0.5) to either more 129- or Cast-monoallelic expression according to which of the two X chromosomes carries the Xist transgene. Thus, Tg-X clones 85, 86, 109, 190 show inactivation of the wild-type Cast/Ei X chromosome (ratio < 0.5), whereas in Tg-X;8 clones 203b, 267, and 339 the 129/Sv X chromosome portion of the X;8 translocation product is inactivated upon doxycycline Xist induction (ratio > 0.5) (Fig. 3c, h). Due to the presence of a *Tsix*-stop allele[32], clone 267 already showed a bias toward Cast-monoallelic gene expression in untreated conditions. Silencing of X-linked genes is slightly reduced when compared to Tg-E clones, but did not differ a lot between Tg-X and Tg-X;8 clones, ranging from 35 to 65% (Fig. 3d–i). This indicated that silencing of X-linked genes induced from Xist transgenes located on X chromosomal and autosomal sequences works equally well.

Next, we tested whether ectopic Xist RNA could silence autosomal genes, focusing on chromosome 12 and 8 gene expression upon *Xist* induction in Tg-12 and Tg-X;8 clones, respectively. Tg-12 clones carry three chromosomes 12, one of 129/Sv, and two of Cast/Ei origin. Therefore, the Cast/all expression ratio is close to 0.66 in doxycycline-untreated clones. This allele-specific ratio shifts in either one or the other direction according to which of the three chromosome 12 carries the transgene. Clone 55 shows the highest efficiency of gene

inactivation with 26% of chromosome 12 genes becoming silenced, whereas in clones 292 and 273 the expression of only 15% of the autosomal genes is affected by *Xist* induction (Fig. 3f).

Similar results were found for Tg-X;8 clones, in which gene expression of the trisomic portion of chromosome 8 shifts from a Cast/all ratio of 0.66 in doxycycline-untreated cells to biallelic gene expression upon Xist induction (ratio < 0.66), with variable efficiency in between different clones (Fig. 3g–i). However, in Tg-X;8 clones 203b, 267, and 339, the inactivation of the autosomal portion of the X;8 translocation product is less pronounced than the X-chromosomal counterpart (Fig. 3i). In fact, only 15–25% of autosomal genes are affected by Xist induction, which is significantly less than observed for *cis*-linked X-chromosomal genes (Fig. 3i). Importantly, chromosome 8 gene expression does not show any change upon *Xist* induction in Tg-X clones, thus excluding any impact of doxycycline treatment on gene expression (Supplementary Fig. 3A). The RNA-seq results were validated by real-time PCR analysis followed by pyrosequencing (Supplementary Fig. 3B, C). Taken together, our findings suggest that ectopic Xist RNA can *cis*-inactivate X-linked genes independently of the locus from which it is induced, even when induced from autosomal sequences. In contrast, Xist's silencing ability of autosomal genes is reduced, suggesting the presence of X-linked sequences or epigenetic features that favor silencing of X-linked genes.

**Xist-mediated silencing upon ESC differentiation**. In all experiments described so far, ESCs were grown in culture media supplemented with leukemia inhibitory factor (LIF) and inhibitors of the MAPK and Gsk3β pathways ("2i" culture conditions), which stabilize the pluripotent ground state[34, 35]. Furthermore, ectopic XCI in pluripotent ESCs relies on constant *Xist* expression and gene inactivation becomes irreversible only upon ESC differentiation[36]. Based on these observations, we asked whether 2i culture conditions and cell differentiation might have an impact on Xist's silencing efficiency. First, we performed a time course experiment in which Tg-E and Tg-12 ESC clones were grown in more "primed" serum + LIF culture conditions supplemented with doxycycline for 6 days. This transfer did not reveal any effect on gene silencing efficiency (Fig. 3j, k; Supplementary Fig. 3D, E). In Tg-E clones, *Rnf12*, *Abcb7*, and *Pgk1* were consistently silenced

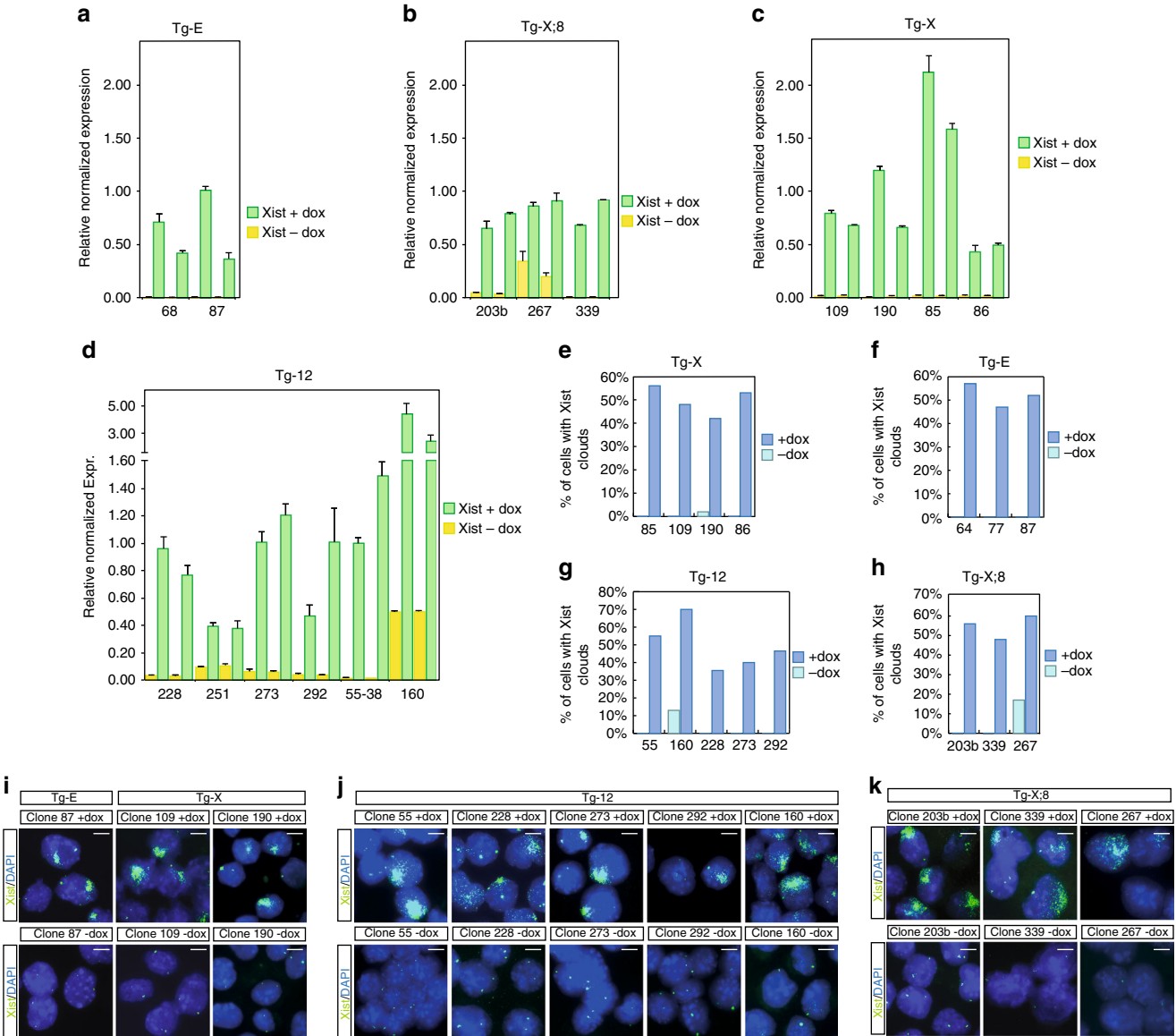

**Fig. 2** Ectopic Xist RNA induction in undifferentiated ES cells. qPCR quantification of Xist RNA in clones **a** Tg-E (68, 87), **b** Tg-X;8 (203b, 267, 339), **c** Tg-X (109, 190, 86, 85), and **d** Tg-12 (228, 251, 273, 292, 55-38, 160) after 5 days of doxycycline treatment. Data of two independent experiments are shown (**e–h**). Quantification of Xist RNA FISH experiments in clones Tg-X (85, 109, 190, 86) **e**, Tg-E (64, 77, 87) **f**, Tg-12 (55, 228, 273, 292, 251) **g**, and Tg-X;8 (203b, 339, 267) **h** after 5 days of doxycycline induction. $n > 100$ nuclei counted per ES clones. **i–k** Representative images of Xist RNA FISH analysis of clones Tg-E (87), Tg-X(109), Tg-X;8 (203b, 339), and Tg-12 (55, 228, 273, 292) after 5 days of doxycycline treatment. Xist, FITC; DNA is stained with DAPI (*blue*). *Scale bars* represent 5 μm

at each tested time point (Fig. 3j; Supplementary Fig. 3D). In Tg-12 clones, the efficiency of gene inactivation is more heterogeneous with clones 55 and 273 showing silencing of three out of four tested genes at each time point of the experiment (*Nampt*, *Tcl1*, and *Pole2*), whereas in clone 228 only *Pole2* and *Tcl1* are affected by *Xist* induction and clone 292 showed very poor silencing for all tested genes (Fig. 3k; Supplementary Fig. 3E).

Next, we differentiated all ESC clones into neurons (Supplementary Fig. 4A, B). Upon neuronal differentiation of Tg-E clones, the X-linked genes *Rnf12* and *Abcb7* showed increased skewing toward monoallelic-129/Sv expression in doxycycline-treated cells compared to ectopic XCI triggered in undifferentiated ESCs (day 0) (Fig. 3l). In contrast, when we followed allele-specific expression of chromosome 12 genes throughout neuronal differentiation of Tg-12 clones, we confirmed silencing efficiencies to be higher for clones 55, 273, and 160 compared to

228 and 292, but silencing efficiency did not increase upon neuronal differentiation (Fig. 3m; Supplementary Fig. 3F). Allele-specific Xist quantitative PCR confirmed that Xist-inducible transgenes remained expressed throughout differentiation in all Tg-12 clones (Supplementary Fig. 3G). However, we cannot exclude that the poor gene silencing observed for clones 292 and 228 might be due to the integration sites of Xist transgenes relative to the few loci we tested in our single-gene silencing assay. Therefore, we performed RNA-seq analysis at day 14 of neuronal differentiation of clones 292 and 55, showing poor and robust Xist-mediated silencing of chromosome 12, respectively (Fig. 3n, o). Chromosome-wide analysis of clone 55 confirmed efficient gene inactivation of chromosome 12 upon Xist induction although the proportion of genes that are affected by Xist did not significantly increase compared to what we observed in undifferentiated ESCs (Fig. 3o, f). Similarly, overall inactivation

of chromosome 12 in clone 292 remained poor in fully differentiated cells with only 10% of autosomal genes becoming inactivated upon Xist induction (Fig. 3o).

We also studied the effects of Xist-mediated silencing upon neuronal differentiation of Tg-X;8 and Tg-X clones, in which the fusion between chromosomes X and 8 results in both partial

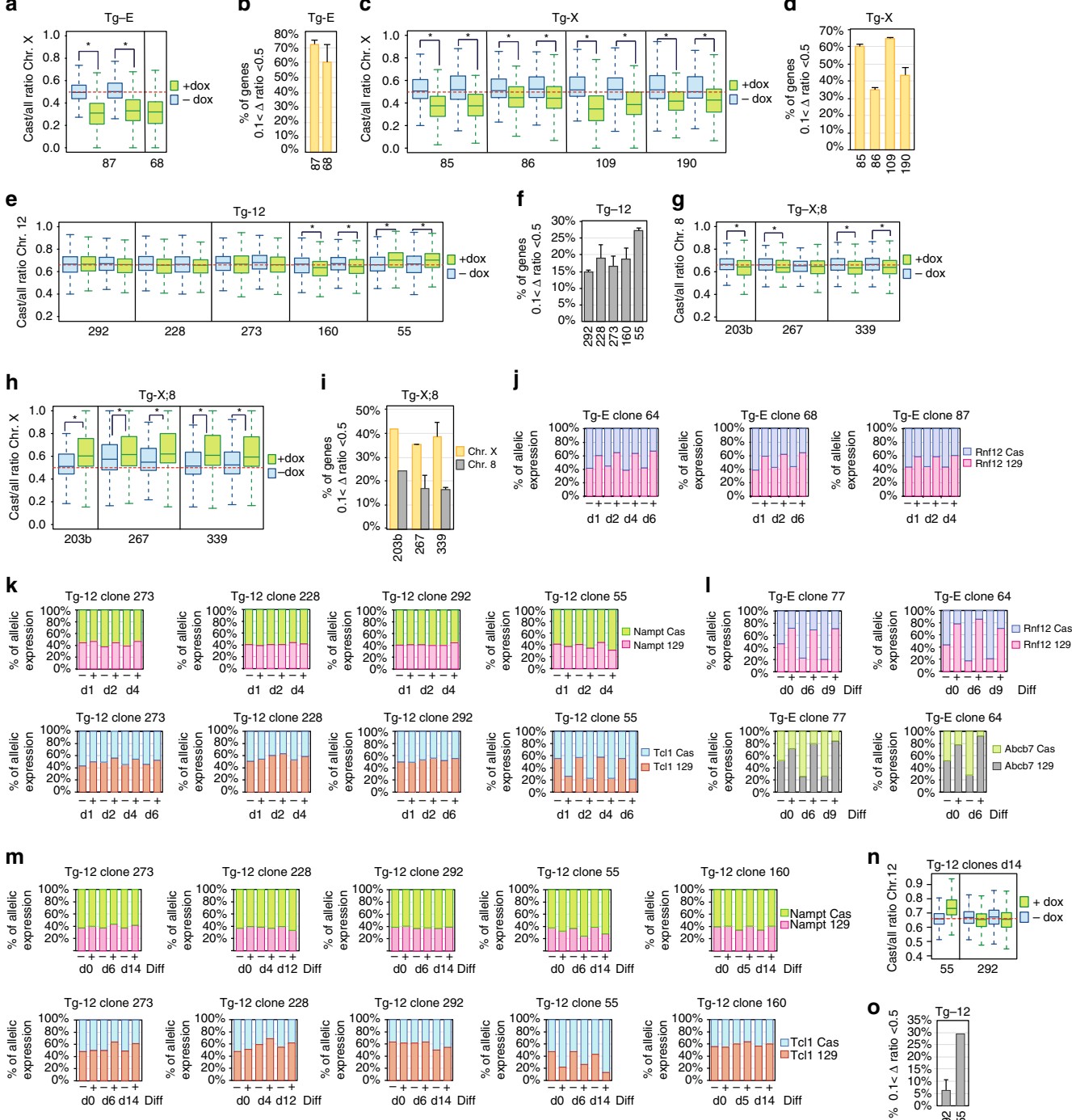

**Fig. 3** Xist-mediated silencing at different genomic loci. **a, c, h** Box plot showing the Cast/all ratio of X-linked genes for Tg-E **a**, Tg-X **c**, and Tg-X;8 **h** clones after 5 days of doxycycline treatment in undifferentiated ESCs. **e, g** Box plot showing the Cast/all ratio of chromosome 12 and 8 genes in Tg-12 and Tg-X;8 clones after 5 days of doxycycline treatment in undifferentiated ESCs. *$p < 0.05$ Mann–Whitney $U$-test. **b, d, f, i** Bar graphs showing the percentage of genes with a Δ ratio between 0.1 and 0.5 after 5 days of doxycycline treatment in undifferentiated ESCs. Δ ratio = (Cast/all ratio (−dox))−(Cast/all ratio (+dox)). For each clone, data from two independent replicates are shown. **j, k** RT-PCR analysis followed by pyrosequencing at different time points after doxycycline induction in undifferentiated ESC clones grown in serum + Lif conditions. Data for *Rnf12* (Chr. X), *Nampt*, and *Tcl1* are shown. **l, m** RT-PCR analysis followed by pyrosequencing at different time points of neuronal differentiation of Tg-12 and Tg-E clones. Data for *Rnf12* and *Abcb7* (Chr. X) and *Nampt* and *Tcl1* (Chr.12) are shown. **n** Box plot showing Cast/all ratio of chromosome 12 genes for Tg-12 clones 55 and 292 upon doxycycline induction at day 14 of neuronal differentiation. **o** Bar graph showing the percentage of genes with a Δ ratio between 0.1 and 0.5 in clones 55 and 292 upon doxycycline induction at day 14 of neuronal differentiation. Δ ratio = (−Cast/all ratio (−dox))−(Cast/all ratio (+dox))

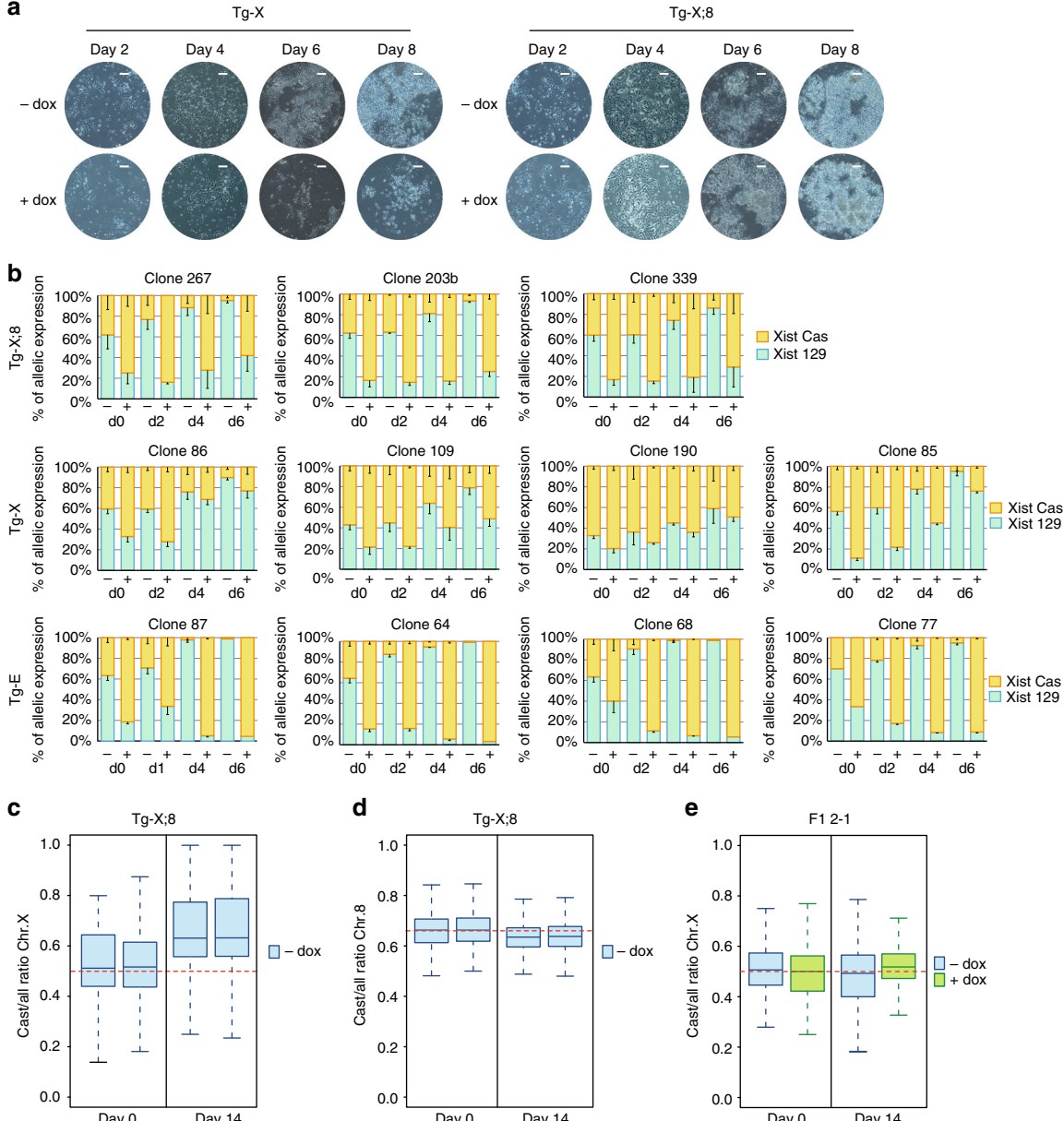

**Fig. 4** Xist-mediated rescue of ESC clones from lethal aneuploidy. **a** Neuronal differentiation of Tg-X;8 and Tg-X clones. Representative bright field images of live cells at day 2, 4, 6, and 8 of neuronal differentiation in doxycycline-treated and -untreated conditions are shown. *Scale bars* represent 100 μm. **b** Xist allele-specific qPCR analysis of Tg-X;8 (*top*), Tg-X (*middle*), and Tg-E (*bottom*) clones upon neuronal differentiation. Day 0 allele-specific analysis refers to total Xist expression levels shown in Supplementary Fig. 5. The mean and SD of three to four independent experiments are shown. **c**, **d** RNA-seq analysis. Box plots showing Cast/all ratio of chromosomes X **c** and chromosome 8 **d** genes in Tg-X;8 clones at day 0 and day 14 of neuronal differentiation in the absence of doxycycline. Data of two independent experiments are shown. **e** RNA-seq analysis. Box plot showing Cast/all ratio of chromosome X genes at day 0 and 14 of neuronal differentiation of wild-type F1 2-1 cells

trisomy of chromosome 8 genes and partial monosomy of X-linked genes (Supplementary Fig. 1C). Forced induction of Xist in Tg-X clones resulted in massive cell death upon differentiation, which was not observed for Tg-X;8 clones, indicating rescue of Tg-X;8 clones by inactivation of the translocated chromosome (Fig. 4a). Indeed, all doxycycline-untreated Tg-X and Tg-X;8 clones showed skewed Xist expression from and inactivation of the 129/Sv X chromosome that is fused to chromosome 8 in the X;8 translocation product (Fig. 4b; Supplementary Fig. 5). This was confirmed by allele-specific RNA-seq analysis of Tg-X;8 neurons that were differentiated in the absence of doxycycline, confirming that Xist-mediated correction of the X;8 chromosomal

rearrangement is crucial for cell survival (Fig. 4c, d). In contrast, wild-type ESCs show random XCI upon neuronal differentiation (Fig. 4e). Since gain of extra chromosomes is usually better tolerated than chromosomal loss[37], silencing of the monosomic X-linked genes on the 129/Sv chromosome rather than partial trisomy of chromosome 8 is most likely responsible for the lethality observed upon differentiation of Tg-X clones.

In conclusion, although ESC differentiation might stabilize Xist-mediated silencing of X-linked genes, as seen upon neuronal differentiation of Tg-E clones (Fig. 3l), it does not affect the variability of Xist's silencing efficiency in between different clones with random integration of Xist transgenes on autosomes.

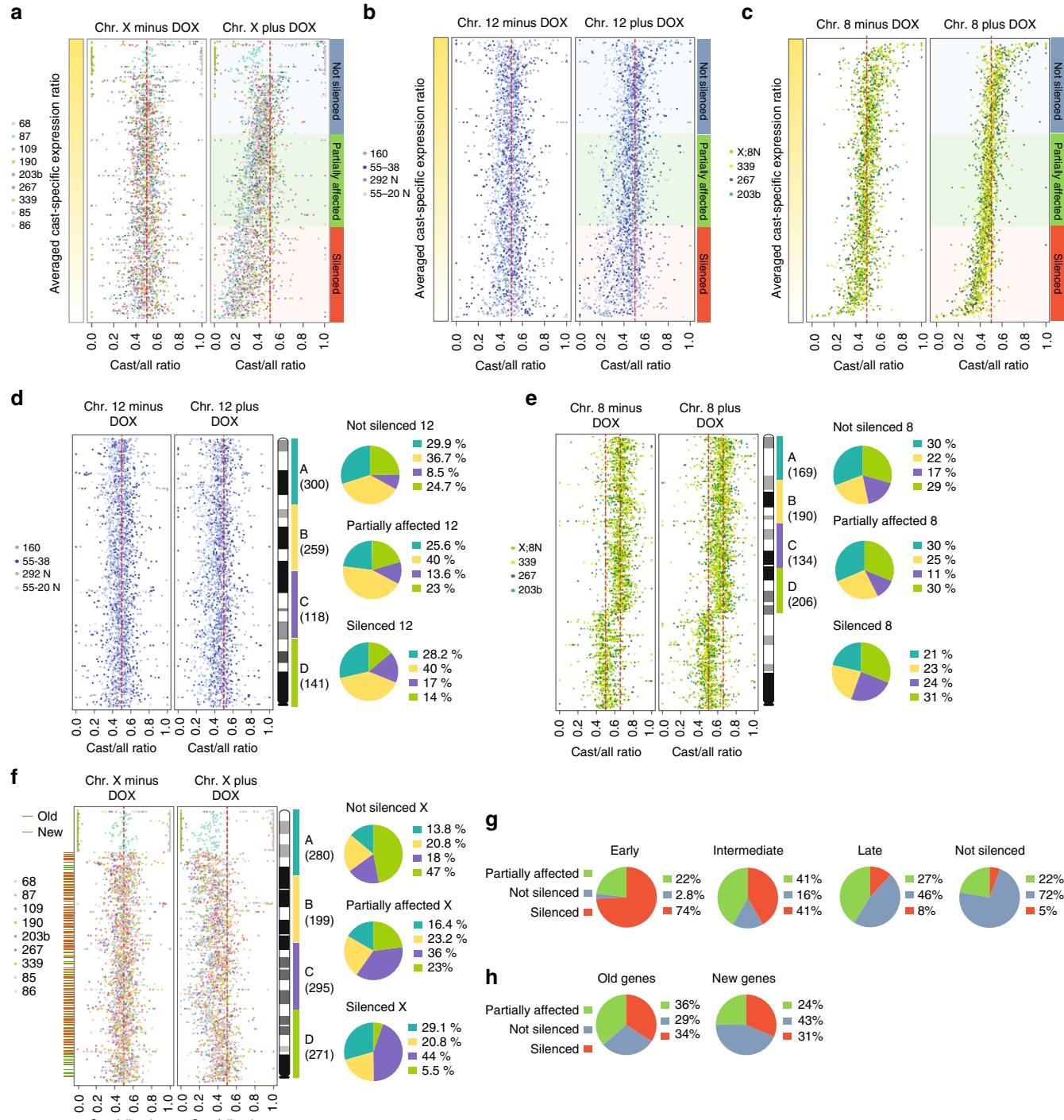

**Fig. 5** Preferential silencing of specific X-linked loci by ectopic Xist RNA. Gene silencing ranking plots for X-linked **a**, chromosome 12 **b**, and chromosome 8 **c** genes. Every *dot* represents the Cast/all expression ratio of a specific gene. About 242 genes are shown in **a**, 351 in **b**, and 336 in **c**. Genes are ranked based on the averaged Cast/all ratio among all clones in each group of clones (Tg-E, Tg-X, Tg-X;8, and Tg-12). Ranked genes are divided in three categories (I) efficiently silenced, (II) partially affected, and (III) not silenced genes. To simplify data visualization, the Cast/all ratios were transformed as follows: (1) For Tg-X;8 clones, carrying Xist transgenes on the 129/sv X chromosome, we used the reciprocal of the Cast/all ratio. Thus, all clones carrying an X-linked Xist transgene behave as if the transgene was integrated on the Cast/Ei X chromosome. (2) For Tg-12 and Tg-X;8 clones, we converted the data from trisomic to disomic by calculating a new Cast/all ratio = (NCast/2)/(NCast/2 + N129). **d**–**f** Genes are ordered by genomic position. Pie graphs show the amount of overlap between the gene categories denoted in **a**–**c** and chromosomal regions A, B, C, and D on chromosomes X, 12, and 8. Gene density along chromosomal regions A, B, C, and D is indicated in brackets. In **f**, *red* and *green lines* represent evolutionary old and new X-linked genes, respectively. **g**, **h** Pie graphs showing the proportion of efficiently silenced, partially affected, and not silenced X-linked genes overlapping with gene clusters that show different inactivation dynamics upon cell differentiation (**g**) and evolutionary old and new X-linked genes (**h**)

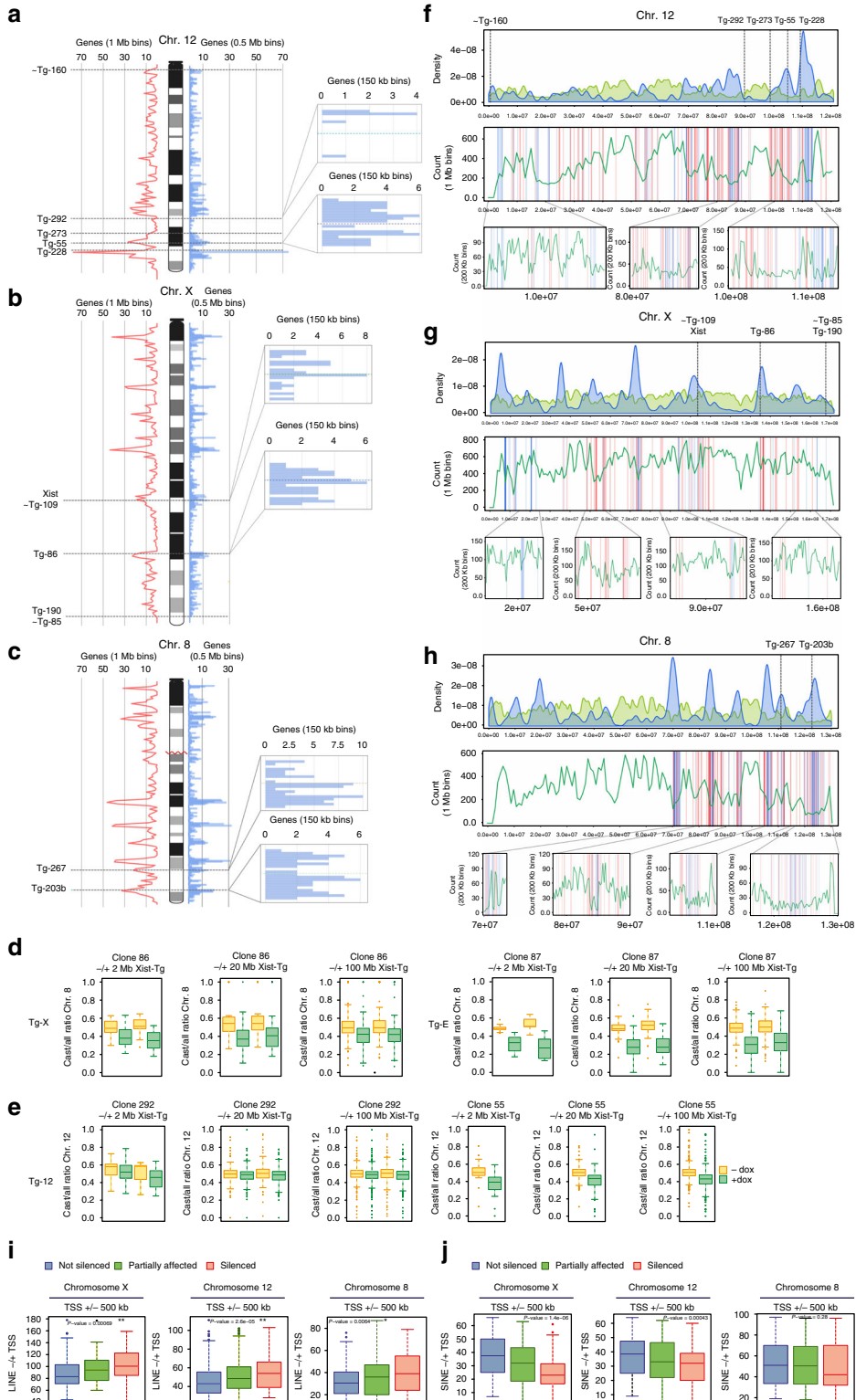

**Fig. 6** LINE and gene density on chromosomes X, 8, and 12. **a–c** Gene density along chromosomes 12 **a**, X **b**, and 8 **c**. *Blue histogram bars* represent 0.5 Mb bins, *red frequency lines* correspond to gene distribution in 1 Mb bins. The integration sites of Xist transgenes are indicated and zoom in of the integration loci is shown. *Blue histogram bars* represent 150 kb bins. **d**, **e** *Box plots* showing the Cast/all ratios of X-linked **d** and chromosome 12 **e** genes in 2, 20, and 100 Mb bins around the Xist transgene integration sites as determined by allele-specific RNA-seq analysis. **f–h** *Top*: LINE density relative to gene density on chromosome 12 **f**, X **g**, and 8 **h**. *Blue*, gene density; *green*, LINE density. *Middle*: LINE distribution along chromosome 12 **f**, X **g**, and 8 **h**, *green frequency lines* correspond to LINE distribution in 1 Mb bins along the chromosomes. *Blue* and *red lines* indicate "not silenced" and "efficiently silenced" genes defined in Fig. 5, respectively. *Bottom*: zoom in of specific loci. *Green frequency lines* correspond to LINE distribution in 200 kb bins. **i** Box plot showing LINE density in 1 Mb bins around the TSS of genes belonging to the three different categories defined in Fig. 5. Data for chromosomes 12, 8, and X are shown. *$p < 0.05$ and **$p < 0.005$ Wilcoxon rank-sum test. **j** as in **i** but for SINE elements

**Preferential inactivation of specific genes on chromosome X.**
Next, we tested whether there are specific genes on chromosomes
X, 8, and 12 that are more prone to inactivation regardless of the
Xist transgene position along the chromosome. To this end, we
evaluated each gene in our allele-specific RNA-seq data sets. We
first calculated the averaged allele-specific expression ratio for
each gene across all clones of each category. Second, we ranked all
X-linked and autosomal genes according to their general degree
of silencing upon doxycycline treatment. Finally, we divided
chromosomes X, 8, and 12 ranked genes into three categories:
(I) genes that are efficiently silenced, (II) genes that are partially
affected and (III) not silenced genes (Fig. 5a–c). Ranking genes in
separated data sets for chromosomes X, 12, and 8 allows us to
relate different silencing effects to genetic and epigenetic features
along each of the three chromosomes, and independently of the
generally more efficient inactivation of chromosome X compared
to autosomes (Fig. 3). Thus, we tested whether its position in a
specific chromosomal region makes a gene more prone to either
becoming inactivated or to escape from Xist inactivation
(Fig. 5d–f; Supplementary Fig. 6). On chromosome 12, silenced,
partially affected, and not silenced genes are evenly distributed
along the entire chromosome length in all tested clones, regard-
less of Xist transgene position. Rather, it correlated with general
gene density (Fig. 5d). Similar results were obtained for chro-
mosome 8, with the exception of a slight tendency of strongly
silenced genes to deviate from the pattern observed for not
silenced and partially affected genes (Fig. 5e). This tendency most
likely reflects more efficient inactivation of chromosome 8 genes
in proximity of the X;8 translocation breakpoint (Fig. 5e).

In contrast, differentially silenced genes are not homoge-
neously distributed along the X chromosome (Fig. 5f). Rather,
X-linked genes are organized in chromosomal blocks that behave
differently in terms of gene inactivation efficiency. Centromeric
genes are more prone to escape ectopic XCI compared to genes
located in the sub-centromeric region of chromosome X,
independently of where the Xist transgene was integrated on
the X chromosome or on the X;8 translocation product (Fig. 5f;
Supplementary Fig. 6A–C). These results indicate a fundamental
difference in the capacity for ectopic inactivation between
X-chromosomal and autosomal genes, and led us to hypothesize
that if XCI is artificially induced in undifferentiated ESC, it always
recapitulates endogenous XCI, independently of the locus on the
X chromosome from which Xist RNA is forced to spread. Indeed,
by comparing the ranked X-linked gene list with clusters of genes
that show different dynamics of silencing upon ESC differentia-
tion[38], we found that 74% of early silenced genes correspond to
strongly silenced genes in our data set and 72% of the genes that
escape XCI overlap with our "not silenced" genes category
(Fig. 5g). Importantly, 95% of the early silenced genes were found
to be at least partially affected in our data set and only 5% of the
reported escaping genes showed any sign of silencing upon
ectopic XCI (Fig. 5g). Next, we addressed the question of whether
the evolutionary features of X-linked genes predispose them to
become inactivated or to escape XCI. X-linked genes can be
classified based on the X chromosome's evolutionary history:
"old" genes are those found on chicken orthologous autosomes,
whereas "new" genes were exclusively added to the mammalian
X chromosome[39–41]. We mapped old and new X-linked genes
on the mouse X chromosome and we found that evolutionary
new genes are enriched in the centromeric region of the X
chromosome that is more prone to escape inactivation (Fig. 5f, h).
Our data therefore confirm previous findings observed for the
human X chromosome[42], indicating that the evolutionary history
of the X chromosome plays a role in establishing the path of
X-linked gene inactivation in XCI in mouse, which appears
independent of the position of Xist on the X chromosome.

**Xist integration sites do not dictate silencing efficiency.**
Depending on the transgene integration locus, chromosome 12
becomes poorly or more efficiently inactivated by ectopic Xist.
Thus, the different degree of chromosome-wide silencing might
rely on specific features of the transgene integration sites. We
therefore compared the Xist integration sites of Tg-12, Tg-X, and
Tg-X;8 clones along chromosomes 12, X, and 8. Among Tg-12,
clones 55 and 292 showed the most efficient and poorest gene
silencing efficiency, respectively, and the genomic environment in
proximity of the transgene integration loci in these two clones
strongly differs in terms of gene density (Fig. 6a). In clone 292,
the Xist transgene was integrated in a gene desert of 2,1 Mb,
whereas clone 55 carries the transgene in a gene-rich chromo-
somal region (Fig. 6a). However, clone 228 is also integrated in a
gene-dense area but does not display more efficient silencing
compared to clones 273 and 160, in which Xist is integrated in
gene-poor areas of chromosome 12. Moreover, when we looked at
the Xist integration sites in Tg-X and Tg-X;8 clones, all clones in
which Xist induction leads to efficient inactivation of X-linked
genes, we found that the Xist transgenes were integrated in
both gene-dense (clones 86, 109, 267, and 203b) and gene-poor
(109, 85) areas of chromosome X and 8 (Fig. 6b, c). These
observations confirm that the presence of X-linked specific
elements, rather than general genomic features such as gene
density, plays a major role in determining the efficiency of
chromosome-wide gene silencing.

Although long-range silencing appeared variable in between
different clones, we also found that genes in 4 Mb regions around
the transgene integration site are always more efficiently
inactivated than genes in 40 and 200 Mb regions (Fig. 6d, e),
regardless of the overall silencing efficiency achieved in different
clones. Thus, clones 55, 86, and 87 efficiently inactivate
chromosomes 12 and X, respectively, but genes in 2 Mb regions
in close proximity of the Xist integration site even showed
stronger inactivation (Fig. 6d, e). Importantly, clone 292
showed poor chromosome-wide inactivation in both ESCs and
fully differentiated neurons but expression analysis of genes
lying in 4 Mb region around the Xist transgene confirmed that
ectopic Xist RNA is capable of inducing local gene silencing
(Fig. 6e).

Thus, ectopic silencing of genes in proximity of the Xist
transcription locus is achieved in all clones, whereas long-range
gene silencing is more efficient for X-linked genes compared to
autosomal genes (Figs. 3, 6d), suggesting that other X-linked
DNA elements are likely to account for the efficient inactivation
of genes that are not in direct proximity of Xist transcription
locus.

**LINEs facilitate silencing of X-linked and autosomal genes.**
Given the fact that LINEs are enriched on the X chromosome[43],
we evaluated the contribution of LINEs to silencing efficiency in
our Xist-inducible system. LINEs are non-LTR retrotransposable
elements that account for up to 19% of the mouse genome[44] and
the majority of which are unable to retrotranspose due to trun-
cation of their 5′ ends. Full-length young LINEs capable of ret-
rotransposition are still present in the mouse genome, although
their exact number remains unclear[45]. Both truncated and full-
length LINEs have been suggested to play a role in XCI; the first
by participating in formation of a silent nuclear compartment
during XCI, and the latter one by promoting the inactivation of
genes that are prone to escape XCI[16]. To test whether truncated
and full-length LINEs facilitate Xist-mediated silencing in our
expression system, we looked at their enrichment in 1 Mb regions
around the TSS of either "efficiently silenced", "partially affected",
and "not silenced" genes along chromosomes X, 12, and 8.

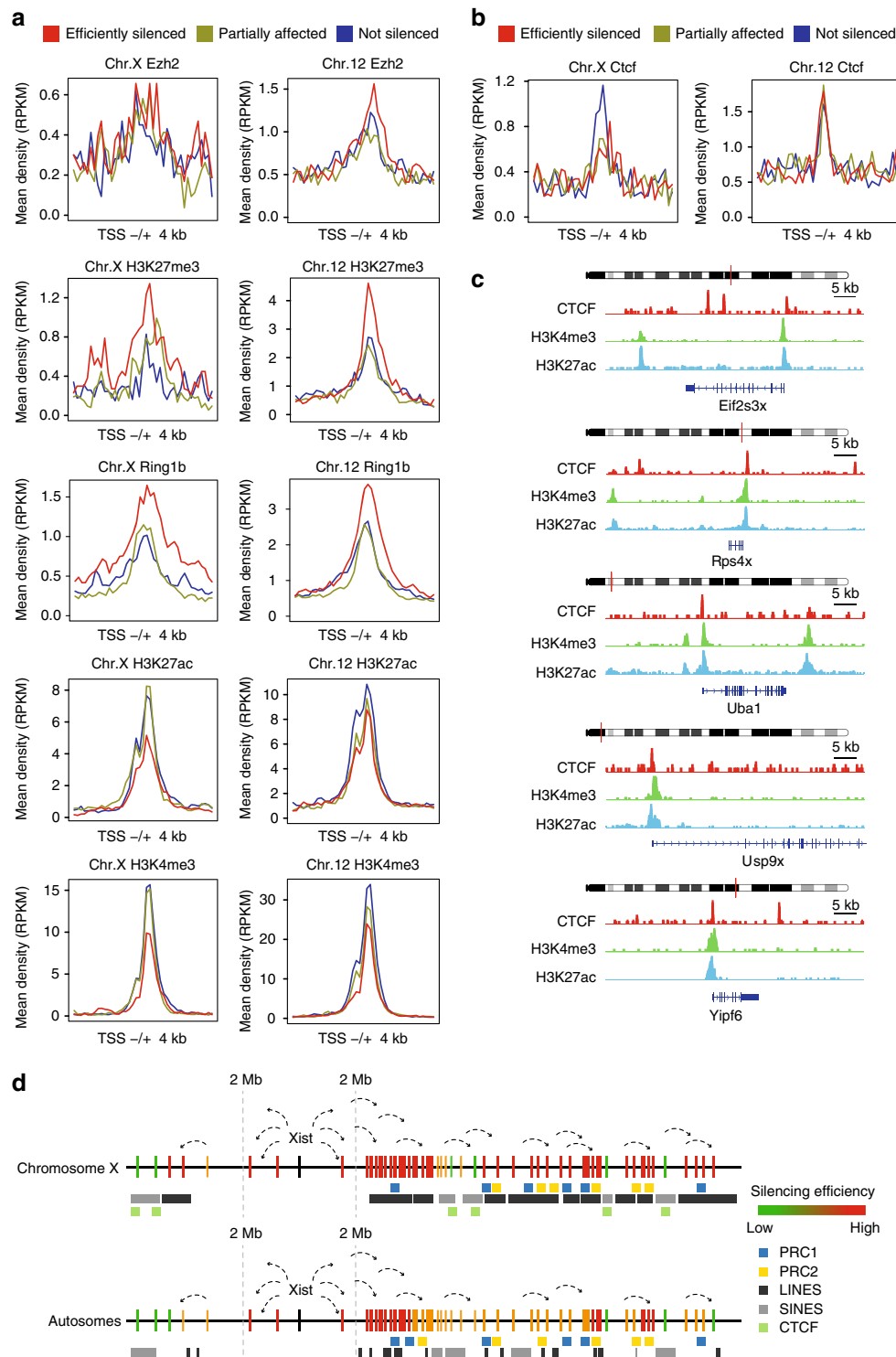

**Fig. 7** Chromatin environment and CTCF enrichment predispose gene silencing and escape. Average density plots for Ezh2, H3K27me3, Ring1b, H3K27ac, H3K4me3 **a**, and CTCF **b** in 8 kb bins around the TSS of X-linked and autosomal genes. **c** ChIP-seq analysis showing CTCF, H3K4me3, and H3K27ac enrichment at the TSS of five not silenced genes. **d** Model for Xist-mediated silencing. Both X-linked and autosomal genes are efficiently silenced in linear proximity of the Xist transcription site, regardless the genomic environment. Long-range gene silencing is more efficient for X-linked genes and relies on LINE elements and PRC1/2 complexes. CTCF sites overlap with SINE elements and CTCF binding mediates X chromosome-specific escape from XCI

This analysis indicated that LINE elements are enriched around the TSS of efficiently silenced genes for all transgenic clones (Fig. 6f–i). In contrast, not silenced genes cluster in LINE-poor regions of chromosomes X, 12, and 8 (Fig. 6f–i). To exclude that this correlation relies on the relative distribution of genes and repetitive elements along the chromosome, we performed the same analysis for short interspersed elements (SINEs), a different class of non-LTR transposable elements that account for 8% of the mouse genome[44]. We did not find any enrichment of SINEs around the TSS of efficiently silenced genes on chromosomes X,

8, and 12 (Fig. 6j). Rather, SINE enrichment correlates with not silenced genes on both chromosomes X and 12 (Fig. 6j).

Since the X chromosome is strongly enriched for LINEs, it is nearly impossible to study Xist spreading from a LINE-poor area of the X (Fig. 6g; Supplementary Fig. 7A). However, in both Tg-12 clone 55 and 8 clones 267 and 203b Xist transgenes landed in LINE-poor regions of chromosome 12 and 8 (Fig. 6f–h; Supplementary Fig. 7) but chromosome-wide silencing is efficiently achieved. Therefore, LINEs are enriched around genes that are efficiently inactivated, but LINE density in close proximity of the Xist transgene integration sites does not affect Xist's spreading efficiency. Since LINE elements are enriched in gene-poor heterochromatic regions of genome, the observed more efficient inactivation of genes in LINE-rich domains might reflect their already pre-repressed status. However, our data indicate that the lower expressed autosomal genes do not correspond to the genes that become more efficiently inactivated upon Xist induction (Supplementary Fig. 7D), thus confirming the specificity of LINEs in facilitating the Xist-mediated silencing of active genes. Taken together, our observations suggest that truncated and full-length LINEs may facilitate Xist-mediated transcriptional inactivation of both X-linked and autosomal genes. In particular, the X chromosome-specific enrichment of LINE elements might explain the efficient long-range silencing of X-linked genes in Tg-X;8 clones in which Xist RNA starts to spread from the autosomal portion of the X;8 translocation product.

**Chromatin environment predisposes gene silencing**. Next, we asked whether the chromatin environment of X-linked and autosomal genes in ESC prior to ectopic Xist induction might predispose a specific gene to be either efficiently silenced or to escape ectopic inactivation. Therefore, we looked at the enrichment of both euchromatic and heterochromatic histone marks around the TSS sites of X-linked and autosomal genes after ranking them based on the degree of inactivation upon doxycycline induction (Fig. 7a). To estimate the density of H3K27me3, EZH2, H3K4me3, H3K27ac, and Ring1b 4 kb upstream and downstream the TSS of (I) efficiently silenced, (II) partially affected, and (III) not silenced genes we used published ChIP-seq data obtained in ESCs[46–50]. Strongly inactivated genes on both chromosomes X and 12 show enrichment of H3K27me3 and Ring1b, and are depleted for active marks such as H3K4me3 and H3K27ac around their TSS prior to inactivation. Interestingly, depletion of H3K27ac seems specific for the X but not for genes located on chromosome 12, suggesting that genes with less active promoters are more likely targeted by the XCI machinery on the X. Examination of average FPKM values of silenced, partially silenced and not silenced genes indeed confirms that efficiently silenced X-linked genes display a significantly lower expression level, which is not observed for efficiently silenced autosomal genes (Supplementary Fig. 7D). Our findings therefore indicate the chromatin environment might be instructive for Xist to function efficiently.

**CTCF is enriched at the TSS of X-linked "not silenced" genes**. To date, several lines of evidence have suggested CTCF to be involved in XCI escape. CTCF is enriched at transition regions between silenced and escaping loci on the Xi[27] and might act to prevent spread of escape into neigbouring regions[26]. Moreover, escaping genes tend to be the only regions that show accessible chromatin on the Xi, and most of these accessible sites correspond to CTCF sites[51]. Thus, we set out to address whether CTCF binding plays a role in mediating XCI escape in our inducible system. Using published data on ESCs CTCF profiles[52],

we assessed the enrichment of CTCF around the TSS of both X-linked and autosomal genes ranked by silencing efficiency upon Xist induction. Our analysis showed enrichment of CTCF at the TSS of "not silenced" X-linked genes relative to partially affected and fully silenced genes. This enrichment was notably absent for autosomal genes escaping XCI, pointing to an X chromosome-specific role for CTCF in mediating escape from XCI (Fig. 7b). Indeed, if active loci were causative of CTCF binding, CTCF would be enriched at the TSS of both X-linked and autosomal not silenced genes. Among the X-linked genes that showed enrichment of CTCF at their TSS, we could detect eight genes that were previously reported to escape XCI. Five of them, Eif2s3x, Yipf6, Uba1, Rps4x, and Usp9x were defined as escaping genes in at least two independent studies[28, 38, 42, 53, 54], and three of them, Usp11, Haus7, and Apoo were reported to escape in one study[55]. These findings suggest a very important role for CTCF in maintaining escape from XCI during evolution of our sex chromosomes.

## Discussion

To investigate the mechanisms of Xist-mediated gene silencing, we developed an inducible Xist expression system in ESCs. Although previous studies have assessed Xist transgenic ESCs, all these approaches were based on a limited number of genomic integration sites, led to functional nullisomy of autosomal genes upon Xist induction, and were performed in non-polymorphic mouse strains[16, 18, 36]. The present study was designed to provide a number of advantages over previous strategies involving Xist transgenes. First, the use of F1 hybrid ESCs derived from 129/Sv and Cast/Ei mouse strains[56] provided a very high density of SNPs and facilitated genome-wide expression analysis by allele-specific RNA-seq[33]. Second, the use of female wild-type and aneuploid ESC lines carrying either an extra copy of chromosome 12 or an unbalanced X;8 translocation allowed us to induce XCI in ESCs without triggering cell death, and to look at its impact during differentiation. Third, by controlling Xist expression in isogenic clones we could directly compare the efficiency of Xist-mediated silencing between sets of ESCs that differ only in terms of Xist transgene integration site on chromosomes X and autosomes.

Our studies indicate that Xist RNA can efficiently spread in cis from different genomic locations, although its ability to trigger gene silencing is position dependent. On the X chromosome, XCI can be triggered from different loci always faithfully recapitulating endogenous XCI: (I) "efficiently silenced" genes highly overlap with genes that become inactivated first upon differentiation of wild-type female ESCs[38], (II) a high overlap was observed between genes that escape XCI[38], and (III) evolutionarily younger genes are more likely to escape XCI than older genes, confirming that XCI efficiency is influenced by the evolution of the X chromosome[42]. On autosomes, the efficiency of Xist's silencing is heterogeneous in between different clones, in line with many X;autosome translocation studies performed in somatic cells[11–13, 57, 58]. Although all tested clones displayed gene silencing in close proximity to the Xist transgene integration site, comparison of the genomic environment of the Xist transgenes indicated variability between transgenes. This variability might be related to gene density, which we find increased in the near vicinity (2 Mb) of the transgene integration site for most transgenes that silence efficiently. Xist's spreading on the X chromosome has previously been proposed to follow a two-step mechanism, initially targeting gene-dense areas[19], with a preference for genes located in spatial proximity to the Xist transcription locus[20]. The variable silencing of autosomal Xist transgenes may therefore be partially explained by local and distal

differences in gene density in conjuction with a favorable spatial landscape for *Xist* to function.

High-resolution maps of Xist RNA localization on the X chromosome showed that LINE-rich domains are not coated by Xist RNA at the onset of XCI[19, 20], however, several studies have reported that LINEs may play a role in XCI[16, 18, 59, 60]. Silent LINEs have been implicated into the assembly of a nuclear heterochromatic compartment into which active genes are recruited upon XCI[3, 16]. Also, LINE elements are enriched in heterochromatic G-dark positive bands[61], and carry H3K9me3 at their promoters[45, 62]. Interestingly, loss of H3K9me3 upon XCI results in decreased gene silencing[63, 64]. Since Xist RNA directly interacts with hnRNP K[65], which is required for H3K9me3 deposition[66], LINEs might aid Xist recruitment into their heterochromatic environment. Here, we have been able to systematically test whether LINEs facilitate gene silencing in the context of different *Xist* transgene integration sites, either on the X chromosome or on autosomes. We found that LINEs are consistently enriched around TSS of efficiently silenced genes. Such an enrichment was not found for SINEs, highlighting the specific role of LINEs in facilitating Xist function. Importantly, all autosomal *Xist* transgenes that led to efficient chromosome-wide inactivation were integrated in LINE-poor areas of chromosome 8 and 12. Thus, LINEs may facilitate the propagation of silencing along the chromosome, rather than working as X-linked-specific binding sites for Xist spreading. Our findings may therefore be explained by assuming that LINE dense X-linked sequences are required to properly form a nuclear heterochromatic compartment, independent of the Xist integration site, then resulting in efficient recruitment of genes for inactivation. This suggests that chromosome 8 and 12, which display an overall reduced LINE density may fail to create a functional nuclear heterochromatic compartment, leading to a reduced efficiency to silence autosomal genes. Importantly, chromosome 8 genes in proximity of the X;8 translocation breakpoint, thus closer to X chromosomal, LINE-rich DNA, appear to be more efficiently silenced than other autosomal genes, which are closer to the Xist transcription locus. This again highlights the role of the X-linked LINE-dense environment in facilitating gene silencing.

Exhaustive analysis of multiple different *Xist* transgenic cell lines indicated that X-linked and autosomal genes that are most efficiently inactivated by Xist tend to show enrichment of both PRC1 and PRC2 components, before *Xist* induction. Accordingly, about half of the X-linked PRC2 sites that are acquired upon ESC differentiation correspond to sites that were marked exclusively by H3K27me3 in undifferentiated ESC[67]. Since PRC2 and PRC1 positively influence each others recruitment[68], and Xist has been reported to directly or indirectly interact with both repressive complexes[65, 69], the observed enrichment of polycomb complexes at the TSS of silenced genes might be involved in Xist RNA or PRC1/2 recruitment, or else in facilitating the Xist-mediated gene silencing process. Interestingly, we find X chromosome-specific depletion of H3K27ac at TSSs of efficiently silenced genes, revealing an epigenetic signature that might facilitate XCI and explain the more efficient inactivation of X-linked genes.

Finally, we found several known escapees in our "not silenced" category of X-linked genes which correspond to genes that always escape XCI regardless of the locus from which Xist starts spreading, i.e., (I) its endogenous locus on the X, (II) several different loci along the wild-type X chromosome, or (III) the autosomal portion of the X;8 translocation product. These findings show that escaping loci have the intrinsic ability to consistently resist XCI, similar to what was previously reported for the *Jarid1c* locus[25]. Although escaping genes have been previously found to co-localize with CTCF binding clusters on the Xi[28], it has been very difficult to address whether CTCF binding itself triggers escape from XCI or whether the transcriptional activity of escaping genes is causative of the CTCF binding. Our results indicate specific enrichment of CTCF at the TSS of X-linked but not autosomal genes that escape ectopic XCI, thus excluding the enrichment of CTCF to simply reflect the transcriptional activity of not silenced genes. Importantly, X chromosome-specific enrichment of CTCF at escaping loci also suggests that maintenance of CTCF binding upon XCI evolution might have allowed clusters of escaping genes to resist the general collapse of topologically-associated domains, a typical feature of the inactive X chromosome[51, 70]. In contrast, autosomal genes have not been selected in favor or against their degree of inactivation upon Xist RNA spreading, thus explaining both the lack of differential CTCF enrichment at the TSS of strongly inactivated and not silent genes, and the less efficient long-range silencing of autosomal genes compared to X-linked genes.

The expansion of CTCF binding sites functionally relies on a repeat-driven mechanism[71]. CTCF motifs were carried to over thousands genomic locations in the mouse genome by transposition of SINE elements[72]. Thus, since CTCF binding prevents both DNA methylation[73] and the establishment of a repressive chromatin environment[74], carrying a CTCF motif within their sequence have conferred SINE elements an evolutionary advantage[71]. Indeed we observed a positive correlation between SINEs and XCI escape, similar to a previous report studying human somatic cells[59]. In the context of XCI, the SINE-driven expansion of CTCF binding sites, together with the enrichment of CTCF at the TSS of escaping genes might therefore explain the positive correlation that we found between SINEs and genes that tend to resist to ectopic Xist induction.

In summary, our data support a model where the X chromosome evolved specific genomic features facilitating efficient XCI and escape of XCI. The accumulation of LINEs facilitated silencing by providing a structural basis for XCI to proceed, guided by a chromatin landscape of genes subjected to XCI, while the X chromosome-specific enrichment of CTCF at escaping loci may enable them to resist Xist-mediated chromosome-wide inactivation (Fig. 7d). In conclusion, the findings we report here, exploring Xist-mediated initiation and spread of XCI from numerous different *Xist* transgenes, provide unique insights into the genomic and epigenomic features that are required to enable Xist RNA-mediated chromosome-wide silencing and local escape.

## Methods

**Recombinant BACs construction**. The X-linked Cast/Ei BAC CH26-171B21 containing the Xist endogenous locus was modified by bacteria-mediated homologous recombination[30]. The pTRE-DsRed-3-5-NEO targeting vector was constructed starting from pTRE-Tight-BI-DsRed2 (Clontech). Homology arms and kanamycin/neomycin resistance cassette flanked by lox sites were amplified by PCR from BAC CH26-171B21 and TOPO-Xist-GFP-NEO using primers listed in Supplementary Table 1. Correctly recombined pTRE-Xist-CH26-171B21 BACs were screened by PCR using primers listed in Supplementary Table 1. Similarly, the M2rtTA transactivator was targeted at the ROSA26 endogenous locus of the 129/Sv BAC RP24-140O11. The M2rtTA-ROSA26-NEO targeting vector was constructed starting from the TOPO-KanaNEO plasmid. 3′ and 5′ homology arms were PCR amplified from BAC RP24-140O11 and ROSA26-m2rtTA-Puro-Amp plasmid using primers listed in Supplementary Table 1. Correctly recombined BACs R26-M2rtTA-RP24-140O11 were screened by PCR using primers listed in Supplementary Table 1.

**ESCs culture and transgenic ESC lines generation**. ESCs were grown either in standard serum + LIF ESC medium conditions containing DMEM, 100 U ml$^{-1}$ penicillin/streptomycin, 20% KnockOut Serum Replacement (Gibco), 0.1 mM NEAA, 0.1 mM 2-mercaptoethanol, 5000 U ml$^{-1}$ LIF, or in feeders-free 2i + LIF conditions supplemented with 1 μM MEK inhibitor PD0325901 (Stemgent) and 3 μM GSK3 inhibitor CH99021 (Stemgent). Transgenic ESC lines were generated using polymorphic F1 2–1 hybrid ESC lines (129/Sv-Cast/Ei)[30]. ESC clones were transfected with 30 μg of recombinant BACs. BAC R26-M2rtTA-RP24-140O11 carrying the reverse tetracycline transactivator M2rtTA was targeted to the 129/Sv chromosome 6 of three F1 2–1 ESC lines (40,XX, 40,XX,t(X;8), 41,XX,dup12). Loss

of a MnlI RFLP upon homologous recombination was used to screen drug-resistant clones for correct targeting events with primers listed in Supplementary Table 1. Similarly, BAC pTRE-Xist-CH26-171B21 was used to target the endogenous locus of the Cast/Ei X chromosome of 40,XX F1 2–1 lines, thus generating the Tg-E clones. Loss of a Tsp509I RFLP was used to screen drug-resistant clones for correct targeting events with primers listed in Supplementary Table 1. Correctly targeted clones were screened by PCR for loss of one X chromosome by a Pf1MI RFLP located in the X-linked gene Atrx using primers listed in Supplementary Table 1. To generate clones Tg-12, Tg-X, and Tg-X;8 F1 2-1 ESC lines 40,XX,t(X;8) and 41, XX,dup12 were transfected with BAC pTRE-Xist-CH26-171B21 and neomycin-resistant clones were screened by DNA FISH. To induce Xist expression, ESC medium was supplemented with 2 µg ml$^{-1}$ doxycycline.

**Fluorescent in situ hybridization**. For DNA FISH, methanol acetic acid fixed cells were dropped on glass slides and incubated at 37° C 24 h. Slides were washed 10 min in 2× SSC buffer at 55 °C (1× SSC: 0.15 M NaCl, 0.015 M sodium citrate), and 5 min in 2× SSC buffer at room temperature before being dehydrate in a gradient of 70, 90, and 100% EtOH. Nick-labeled DNA probes (DIG or BIO Nick-translation kit, Roche) were dissolved in hybridization mixture (50% formamide, 10% dextrane, 2× SSC, pH = 7.5) and 100 ng µl$^{-1}$ mouse Cot-1 DNA (Thermo Fisher Scientific) to a final concentration of 1 ng µl$^{-1}$. The probe mixture was applied to the cells, covered with a glass coverslip, incubated 3′ at 75° C and let to cool down for 30 min on the heating plate after having turned it off. Slides were then incubated overnight at 37° C in a humid chamber filled with 50% formamide in 2× SSC buffer. After hybridization, slides were washed for 10 min in 2× SSC buffer at room temperature, two times 10 min in 0.1× SSC buffer at 55° C and 10 min in low-salt buffer at room temperature (100 mM Tris, 150 mM NaCl, 0.05% Tween). Detection was done by incubation with FITC-labeled anti-digoxigenin antibody (Roche, 11207741910, 1:100) and Alexa594-labeled Streptavidin (Thermo Fisher Scientific, S11227, 1:100) in low-salt buffer containing 1% of low-fat milk for 60 min at 37° C. Slides were washed for 10 min in low-salt buffer and mounted with ProLong Gold Antifade with DAPI (Molecular Probes). The following BACs were used as probes: CH26171B21 (Chr. X), RP23477B14 (Chr. 8), and RP24112A14 (Chr. 12).

For Xist RNA FISH, ESCs were fixed for 10 min with 4% paraformaldehyde (PFA)–PBS at room temperature, washed with 70% EtOH, permeabilized 4 min with 0.2% pepsin at 37 °C and post-fixed with 4% PFA–PBS for 5 min at room temperature. Slides were washed twice with PBS and dehydrated in a gradient of 70, 90, and 100% EtOH. The Xist probe was a 5.5 kb BglII cDNA fragment covering Xist exon 3–7. The probe was dig-labeled (DIG Nick-Translation Kit, Roche) and dissolved in hybridization mixture (50% formamide, 2× SSC, 50 mM phosphate buffer (pH 7.0), 10% dextran sulfate) and 100 ng µl$^{-1}$ mouse Cot-1 DNA (Thermo Fisher Scientific) to a final concentration of 1 ng µl$^{-1}$. After 5 min of denaturation, the probe was pre-hybridized for 45 min at 37° C, and slides were incubated in a humid chamber filled with 50% formamide in 2× SSC buffer at 37° C overnight. After hybridization, slides were washed once in 2× SSC, three times in 50% formamide-2× SSC, both at 37° C and twice in TST (0.1 M Tris, 0.15 M NaCl, 0.05% Tween 20) at room temperature. Blocking was done in BSA–TST for 30 min at room temperature. Detection was done by subsequent steps of incubation with anti-digoxigenin (Roche 11093274910, 1:500) and two FITC-labeled antibodies (Roche 31627, 65-6111 1:250) in blocking buffer for 30 min at room temperature. Coverslips were washed twice with TST between detection steps and once finally with TS (0.1 M Tris, 0.15 M NaCl). Dehydrated coverslips were mounted with ProLong Gold Antifade with DAPI (Molecular Probes).

**ESCs differentiation**. ESCs grown in conventional serum + LIF conditions were pre-plated on cell culture dishes for 40 min and then seeded on feeders-free gelatin-coated culture dishes containing EB differentiation medium (IMDM-glutamax, 15% fetal calf serum, 100 U ml$^{-1}$ penicillin/streptomycin, 0.1 mM NEAA, and 50 µg ml$^{-1}$ ascorbic acid). During differentiation, the culture medium was refreshed daily. To induce neuronal differentiation, ESCs grown in feeders-free 2i conditions were seeded at a density of 10,000 cells per cm$^2$ on 10 µg ml$^{-1}$ laminin-coated (Sigma Aldrich, L2020) dishes in neuronal differentiation medium (50% Neurobasal (Gibco), 50% DMEM:F12 (Gibco), 100 U ml$^{-1}$ penicillin/streptomycin, 1% N-2 supplement (Gibco 17502-048), 2% B27 supplement (Gibco 17504-044), 2 mM L-Glutamine (Gibco 25030081), and 0.05 mM 2-mercaptoethanol (Gibco 31350010). Cells were refreshed daily and neuronal medium was supplemented with 250 nM retinoic acid between day 4 and day 8 of differentiation. After 8 days, cells were split at low density and further cultured in neuronal medium to obtain fully differentiated neurons around day 12–14 of differentiation.

**Expression analysis**. Cells were lysed by direct addiction of 500 µg of TRIZOL and total RNA was extracted according to the manufacturer's instructions (Invitrogen). To remove genomic DNA contamination, samples were treated for 15 min at 37° C with DNaseI (Invitrogen). Next, 1 µg of RNA was reverse transcribed by Superscript II reverse transcriptase with random hexamers (Invitrogen). For quantitative PCR and allele-specific quantitative PCR, gene expression levels were quantified using 2× SYBR Green PCR Master Mix (Applied Biosystems) in a CFX384 Real-Time machine (Bio-Rad) with primers listed in Supplementary Table 2. Expression levels were normalized to actin b using the $\Delta C_T$ method. To perform

pyrosequencing, oligos for real-time PCR amplification and sequencing were designed according to the PyroMark Assay Design software guidelines and are listed in Supplementary Table 4. Successfully amplified PCR products were purified and annealed with the sequencing primer for pyrosequencing using the PyroMark Q24 (Qiagen).

**Targeted locus amplification analysis**. In brief, ES cells were crosslinked using formaldehyde and DNA was digested with NlaIII. The samples were ligated, crosslinks reversed, and the DNA purified. To obtain circular chimeric DNA molecules for PCR amplification, the DNA molecules were trimmed with NspI and ligated at a DNA concentration of 5 ng µl$^{-1}$ to promote intramolecular ligation. After ligation, the DNA was purified, and six 25 µl PCR reactions, each containing 100 ng template, were pooled for sequencing. All used oligos are listed in Supplementary Table 3. PCR products were NGS library prepped using Illumina NexteraXT NGS library preparation according to manufacturer's protocols. We performed sequencing of targeted locus amplification libraries on the Illumina MiSeq platform pooling ~20 libraries per V2 PE150 sequencing run. Reads were mapped using split-read aware alignment with BWA mapping software.

**Density plots**. Available ChIP-seq data sets were retrieved from the Gene Expression Omnibus (GEO) database and mapped to reference genome mm10 as previously described[75], i.e., sequences with low complexity that are unlikely to map uniquely to the genome were removed from the described ChIP-seq data sets using prinseq-lite with the dust method with 7 as threshold. The remaining sequences with a Phred score < 70 were mapped to the mm10 reference genome using Bowtie v0.12.7, where a seed length of 36 was used in which we allowed a maximum of two mismatches. If a read had multiple alignments, only the best matching read was reported. Duplicated reads were removed. ChIP-seq data sets with multiple replicates were merged. For the ChIP-seq density plots mapped reads were counted in bins of 200 bp in 8 kb centered on the TSSs of genes. Reads were normalized to RPKM. Genes were divided in categories and average RPKMs were plotted.

**Box plots**. The positions of all LINE and SINE elements along the entire length of chromosomes X, 12, and 8 were obtained from the mm10 repeat masker table (UCSC genomic browser). LINE and SINE elements were counted in 1 Mb regions centered on the TSSs of genes. Genes were divided in categories and plotted in box plots.

**RNA sequencing**. RNA samples were prepared with the Truseq RNA kit, sequenced according to the Illumina TruSeq v3 protocol on the HiSeq2000 with a single 43 bp read and 7 bp index. Allele-specific analysis was performed as previously described with minor modifications[33]. Mouse Cast/Ei and 129/Sv genomes were built from the Mouse Sanger database project (version 3) and Mouse mm10 reference genome. Raw sequencing reads were aligned to both genomes with the TopHat aligner (v2.0.6). RefSeq gene annotation from UCSC genome browser was used as the reference for known transcripts. For each sample, mapping results against both parental genomes were merged to keep the best alignment for each read. Only unique alignments were reported for downstream analysis. At each SNP site, the number of bases matching each genotype was counted to obtain the abundance of the 129 and Cast alleles (samtools mpileup v0.1.19). The allelic counts per gene were then estimated by summing up the number of 129- and Cast-specific reads of all exonic SNPs within the gene. Genes containing a single-polymorphic site were included in the analysis only when they show a coverage higher than eight reads, whereas for genes carrying multiple informative SNPs the coverage threshold was of five reads per polymorphic site.

**Data availability**. All sequencing data that support the findings of this study have been deposited in the National Centre for Biotechnology Information GEO and are accessible through the GEO series accession number GSE92894. All other data that support the findings of this study are available from the corresponding authors upon reasonable request.

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

## Acknowledgements

This work was supported by grants from the Netherlands Organization for Scientific Research (NWO-VICI 865.10.003) and the European Research Council (ERC 260587) to J.G.; and by an EMBO short-term fellowship (ASTF640-2014) to A.L. We are thankful to Sarra Merzouk for carefully reading the manuscript and Aristea Magaraki and Federica Federici for help with some of the experiments.

## Author contributions

A.L. and J.G. conceived and designed the study. A.L. performed all the experiments with the help of F.L. and A.A., I.V. pre-processed the RNA-seq data sets assisted by N.S., J.H. B. performed data analysis with A.L., E.S. performed the TLA experiments, A.L. wrote the manuscript with J.G. with input from E.H. and R.A.P.

## Additional information

**Competing interests:** The authors declare no competing financial interests.

