## [Peer Review File · Nature Communications]

Reviewers' comments:

Reviewer #1 (Remarks to the Author):

In this study Loda et al. use ectopic expression of Xist RNA from X chromosomal and autosomal locations to study the characteristics associated with Xist spreading and efficiency of gene silencing. Interestingly, they observed that ectopic expression from any X chromosomal location was able to efficiently induce X inactivation while if the transgene was inserted on an autosomal location then silencing was less efficient and more importantly relied on the site of insertion on the autosome. Authors also observed that insertions in gene rich region were more efficient than gene poor regions, indicating that gene density positively contributes towards Xist function. Authors also propose that in gene poor regions presence of LINE elements could facilitate silencing although it is not clear how this could be achieved. Authors also found a correlation with CTCF binding on X chromosomal sites but not on autosomal sites suggesting that chromatin architecture may also contribute towards the silencing mechanism.

-Overall, I found it an interesting study using hybrid cell lines and inducible ectopic Xist transgenes to study the mechanism of silencing. The work is of high quality and experiments are well-controlled. I am therefore supportive of publication. However, I am concerned that the manuscript is very descriptive with several correlations that have been observed before and falls short of really understanding why for example gene rich regions facilitate Xist mediated silencing and how LINE elements contribute towards this process. Authors don't really provide the reader with a plausible explanation of why this could be happening. I would therefore recommend that authors should at least make an effort in the discussion to elaborate on these aspects.

-Since authors have previously performed allele specific HiC, they could also explore further the chromatin interaction landscape with respect to the transgene insertions. It was surprising that this aspect was not explored at all in the manuscript, which would make the manuscript more interesting.

-Authors should proof read the manuscript for spelling mistakes.

Reviewer #2 (Remarks to the Author):

In this manuscript, Loda et al. use an elegant transgenic approach to investigate the determinant of Xist-mediated silencing in the mouse. The strength and originality of their strategy relies (i) in the parallel analysis of multiple integration sites for inducible Xist, on the X and on autosomes, (ii) in the use of hybrid cells which allows allelic expression analysis, and (iii) in the deepness/extensiveness of the downstream analyses. In particular they use RNA-seq to precisely measure the extent of Xist induced silencing, which, combined to the characterization and comparison of multiple integration sites, allow the authors to assess the correlation between silencing efficiency and genomic and epigenomic features.

The authors provide strong argument to conclude that Xist silencing is much more efficient on the X than on autosomes (even though only 2 autosomes were tested). The analysis of Tg-X;8 clones is in particular very informative as it clearly shows the difference in silencing efficiency between chromosome X or 8 portion. The authors also provide convincing evidence supporting that silencing efficiency is linked to (i) the chromosomal history of the X chromosome, (ii) gene density around Xist integration site (iii) LINES and SINES density and (iv) chromatin environment around TSS. This represents a significant advance compared to previous analyses of Xist silencing capacities in ectopic versus endogenous conditions.

Major comments

1. Assessment of allelic expression by RNA-seq convincingly shows more robust silencing when

Xist is on the X chromosome than on autosomes. Yet, statistical testing could be implemented to validate the data further. There is also some degree of variability between clones that is not discussed. In particular, Xist induction in Tg-X clone 86 doesn't appear to lead to massive silencing as the Cas/all ratio barely decreases (Figure 3B). However the percentage of Xist positive cells in this clone is similar to other clones (Figure 2E), and in this clone the integration occurred in a gene-dense area (Figure 6). More importantly, unlike what is stated line 154, the RFLP RT-PCR analysis do not convincingly support the RNA-seq data (in addition panels shown in Fig. S3 are barely readable): in Figure S3G the allelic bias for G6pd upon Dox treatment is limited and not much stronger than for chr12 genes in Tg-12 clones shown in S3J. Similarly, and in contrast to what the authors conclude (line 168), RFLP analysis performed on cells grown in serum+LIF doesn't reveal a clear shift in the allelic expression (Figure 3F). Strikingly, there is a strong allelic bias (~80%) for MeCP2 and G6pd (albeit to a lesser extent) toward the Cas allele in untreated cells; In Tg-12 clones, the Chr12 gene Fcf1 is more expressed from the 129 allele despite the presence of 2 Cas Chr 12. Altogether, these observations raise question as to the conclusiveness of the RFLP analyses.

2. The section "Xist-mediated inactivation of the X;8 translocation product rescues ESC clones from lethal aneuploidy" doesn't lead to an essential conclusion as such, compared to the rest of the manuscript and blocks to some extent the logical progression of the reading. The fact that inactivation of the translocated product is required for survival of cells with unbalanced translocation has been already described (although I acknowledge that the inducible system used here allows to experimentally test this hypothesis). Nevertheless, I would suggest to remove this section, while keeping the allelic analysis of Xist expression upon dox treatment and differentiation in the context of the previous section ("Xist-mediated gene inactivation efficiency is locus dependent"); this would nicely link the allelism in Xist expression to the silencing. In addition, as this previous section is already a bit long, it could be split in two with a new section starting with the neuronal differentiation of transgenic cells (from line 170).

3. Figure 6: The link between degree of silencing and gene density around Xist insertion site is supported by the comparison of Tg-12 clones 55 and 292. However, more clones should be tested; according to this hypothesis, in Tg-12 clones 228 and 273, Xist should be integrated in a gene poor region, whereas in Tg-12 clone 160 gene density around Xist should be higher. Furthermore, the authors later refer to Tg-X clones 86 and 190 and Tg-X;8 clones 267 and 203b as showing efficient inactivation of X-linked genes. However, Tg-X clones 85 and 109 show more efficient silencing than 86 and 190 (Figure 3B); it would thus be interesting to also know their integration sites. Gene density in 1 Mb bins in panels A-C should be plotted to the same scale for proper comparison.

Minor comments

Line 85: wrong reference to figure panels, should be Figure 1D-1E

Line 97-99 and Figure 2: The author should explain why they used ESC carrying a Tsix stop signal

Figure 2 I-K: the Xist RNA cloud appears much larger than normal, endogenous clouds. Did the authors somehow assess this?

Line 167: Figures 3H-3I: should read 3F-3G

Figure 3I, upper panel; Figure 3J, Figure 4C: description of the x axis is missing (dox/diff)

Figure 4S, 5S, 6S: should be S4, S5, S6 in the figure and in the text (line 205).

Line 207-208: "...percentage of cells that show a Xist-coated chromosome..."

Figure 5: Could the authors define what each category correspond to, in terms of allelic ratio? In addition, the sentence "On chromosome 12, the distribution of silenced, partially affected and not silenced genes was invariant along the entire chromosome length regardless of Xist transgene position" (line 235-236) may be misleading, as for X-linked genes, the distribution is also similar ("invariant") in all clones. It is more that, on chromosomes 12 (and 8), silenced, partially affected and not silenced genes are evenly distributed along the chromosome in all clones, without forming blocks?

Line 389: Misspelling of systematically.

Reviewer #3 (Remarks to the Author):

Gribnau and colleagues investigate the efficiency of Xist RNA-mediated silencing from autosomal and X-linked integration sites of a dox-inducible Xist transgene (Tg) in mouse ESCs. For their study, the authors selected clones with Xist-Tg integrations on: 1) a copy of the trisomic chromosome 12 (Tg-12), 2) one of two X-chromosomes in female cells (Tg-E, at the endogenous Xist locus, or Tg-X, distal from it), and 3) an unbalanced X;8 translocation (Tg-X;8). Although several other groups previously examined the efficiency of Xist spreading and silencing in transgene integrations, as well as balanced X-autosome translocations, the extent and reach of Xist-mediated gene silencing was subject to expression monosomy: as a result, silencing of putative haploinsufficient autosomal genes could select against Tg integrations with strong Xist function. The key aspect of the Loda, et al. manuscript is that expression monosomy of chr8 and chr12 genes is avoided, allowing the authors to compare the relative efficiencies of Xist silencing on X-linked vs. autosomal chromatin.

The scientific premise of this study is strong and represents a significant advance over previous studies. Likewise, the primary technical approach (allele-specific RNA-seq in F1 hybrid ESC cell lines of 129 x Cast parental origin) is sensible, and the mapping of the Tg integration sites is also a real strength of this study. However, insufficient details in the in the figure legends and methods sections confound interpretation of the RNA-seq results, and figures show overly condensed aggregated data that are difficult to appreciate. In addition, gene-specific approaches towards measuring allelic expression ratios are quite clearly flawed (RFLP assay), and should be either redone or replaced by qRT-PCR using allele-specific primer pairs. Additional minor errors in the presentation of this manuscript should be addressed as well. Although interpretation of these results is muddled by these issues, the study represents a strong approach to a fundamental problem in the field.

Major issues:

- 1) Given the large number of ESC clones, readers would benefit greatly from a reference table that goes beyond Fig. 1a, and summarizes karyotype, Tg-integration coordinates (from TLA) including allele (129/Cast), and additional information (e.g. presence of Tsix Stop allele). This supplementary table should also list the GEO sample accession number(s) for each clone.
- 2) RFLPs after PCR are used for both clone screening and measuring allelic ratios of expression, but it's clear from the sub-optimal digestion patterns seen in Fig. 1c, S1 and S3, that PCR reactions had plateaued. Once reactions run out of primers/dNTPs, template strands are no longer copied but reanneal with reverse complement template strands. These can originate from the other allele, producing a mismatch at the polymorphic site that is resistant to restriction enzyme digestion. The over-abundance of undigested PCR product in the gel images is likely due to this effect, and compromises most of Fig. 3. Options are to repeat RFLP assay from qPCR samples (during exponential phase, or with a final extension in a fresh PCR reaction), or to perform qPCR with allele-specific primer pairs (as done for Xist in Fig. 3j and 4c).
- 3) The RNA-seq data would be much more informative if discussed right away in the context of each integration site. Much of figures 3 and 4 could move into the supplement, which would allow for more extensive presentation of the RNA-seq data, and their correlation analyses (currently Figs. 5,6). Superimposition of the data in Fig. 5 make it largely unreadable. Non-aggregated data from different clones (as in Fig. S6) could be presented at different distances from the integration site (1-2 Mb, 20 Mb, and chromosome-wide).
- 4) Were allelic ratios calculated by summing over all genic variants or only exonic ones? Were repetitively aligning reads and transcripts excluded? The methods section refers to a previous study, but that one didn't aggregate allelic ratios across the chromosome. More detail on the bioinformatics is necessary. Plotting allelic ratios of RNA-seq against allele-specific qPCR (or redone RFLP) for a set of integration-proximal and -distal genes would be helpful.

5) The LINE/gene density correlations need to be checked (Fig. 6), and the figure legend is insufficient to allow interpretation. Panels F,G,H are almost certainly incorrect (either the binning/smoothing is excessive, or a script error – compare to Fig. S7). The figure makes a strong point that Xist silencing can spread long-range from gene-rich regions (which by definition are LINE-poor). Conversely, gene-poor regions are enriched in LINES and more repressed by default. The authors argue that silencing of genes in such LINE-rich domains is more efficient, but that's likely due to their already pre-repressed status. A panel showing estimated expression (e.g. FPKM x allelic ratio) of genes from LINE-poor vs. LINE-rich regions (+/- dox) would be required to support the authors claim.

Minor points:

- 1) The presence of the Tsp509I site on the Cast allele in the first diagram (Fig. 1a) is inadvertently misleading. This restriction site is specific to the 129 allele.
- 2) The y-axis in the qPCR quantification data is unclear – is it directly relative to Actin B? Where does Xist expression from its endogenous promoter fall on this scale? Fig. S2A is unclear whether Xist induction is plotted against Actin B or endogenous Xist levels (as alluded to on pg. 3). Fixing the scale would also enable comparison across panels.
- 3) Have the authors tried to ascertain Tg copy numbers from the TLA data? Do they correlate with Xist induction?
- 4) Figure legends are insufficient throughout, including Fig. 4a,b. Also, why is the skewing in non-induced Tg-X;8 clones incomplete (Fig. 4d,e), if dox-induced lethality is due to Xist silencing of the monosomic 15 Mb?
- 5) Pie graphs in Fig. 5d-f should also list gene density (or expressed gene density) across the domains A,B,C,D for comparison.
- 6) What about trans-effects? Are there differentially expressed genes on non-Tg bearing chromosomes in dox? The referenced trisomy 21 paper (human XIST Tg) showed significant off-target effects.
- 7) If the authors stick with RFLP analysis, these primer pairs should be listed in the supplement. TLA primers are also missing as well.

We would like to thank the reviewers for the suggestions made to improve our manuscript. We are also very excited that we have been able to answer most of the questions raised by the reviewers, and think that the implemented changes have improved our manuscript substantially. Our answers to the specific comments made by the reviewers are listed below and all changes are highlighted in red in the manuscript text file.

Reviewer #1 (Remarks to the Author):

-Overall, I found it an interesting study using hybrid cell lines and inducible ectopic Xist transgenes to study the mechanism of silencing. The work is of high quality and experiments are well-controlled. I am therefore supportive of publication. However, I am concerned that the manuscript is very descriptive with several correlations that have been observed before and falls short of really understanding why for example gene rich regions facilitate Xist mediated silencing and how LINE elements contribute towards this process. Authors don't really provide the reader with a plausible explanation of why this could be happening. I would therefore recommend that authors should at least make an effort in the discussion to elaborate on these aspects.

As suggested by the reviewer we have more thoroughly discussed our findings in a functional context. With respect to gene rich regions in relation to silencing we have now included the exact location of the integration sites of nearly all Xist transgenes studied, and have to conclude that, although there is variegation of cis silencing in between clones, X linked genes are more efficiently silenced than autosomal genes. In the best controlled setting where Xist has been integrated on the autosomal part of the X;8 translocation product, this is most clear. This indicates the presence of elements, chromatin modifications, or other features that facilitate Xist silencing, and in the revised version of the discussion of this manuscript we discuss a potential role for LINEs that might explain this finding. We have added a paragraph where we discuss the previous findings that LINEs form a heterochromatic compartment that subsequently recruits genes to be silenced (Chaumeil et al G&D 2006, and Chow et al., Cell 2010). As the LINE density on 8 and 12 is much lower than on the X, lack or less efficient formation of such a core might make it more difficult to recruit autosomal genes. In addition, we found that the H3K27Ac is reduced for efficiently silenced genes on the X, which is much less pronounced on autosomes. This suggests that efficiently targeted genes possess less active TSSs, which may be a feature that co evolved with the evolution of the sex chromosomes. We have also added this point to the revised discussion.

-Since authors have previously performed allele specific HiC, they could also explore further the chromatin interaction landscape with respect to the transgene insertions. It was surprising that this aspect was not explored at all in the manuscript, which would make the manuscript more interesting.

We agree with the reviewer that this is a hypothesis that we would really like to test, but unfortunately the sequencing depth was not deep enough to address this question properly. At the moment one complete project is devoted to answer this question as it requires lot of effort and skills, and we apologize to this reviewer that we will not be

able to conclude this study in the near foreseeable future to include it in the revised manuscript.

-Authors should proof read the manuscript for spelling mistakes.

We did as suggested.

Reviewer #2 (Remarks to the Author):

Major comments

1. Assessment of allelic expression by RNA-seq convincingly shows more robust silencing when Xist is on the X chromosome than on autosomes. Yet, statistical testing could be implemented to validate the data further.

We thank the reviewer for his positive comments, and as suggested we have performed the Mann-Whitney U test to compare silencing in Tg-E, Tg-X, Tg-X;8 and Tg12 clones (implemented in revised Figure 3). This analysis confirms that silencing is significant for all X linked sequences in Tg-E, Tg-X, and Tg-X;8 clones. For Tg-12 only clones 55 and 160 show significant silencing, whereas most chromosome 8 genes on the X;8 translocation product show significant silencing.

There is also some degree of variability between clones that is not discussed. In particular, Xist induction in Tg-X clone 86 doesn't appear to lead to massive silencing as the Cas/all ratio barely decreases (Figure 3B). However the percentage of Xist positive cells in this clone is similar to other clones (Figure 2E), and in this clone the integration occurred in a gene-dense area (Figure 6).

We agree with the reviewer that Xist mediated silencing of X-linked sequences appears variable in our clones with random Xist transgene integrations, but even in clone 85 gene silencing is more robust than silencing of autosomal genes in any of the other clones. In the revised discussion of our manuscript we discuss the observed variability in the context of the integration site of our transgenes. Close examination of silencing efficiency in relation to more integration sites that we determined, indicate that gene density might explain some but not all of the observed variability, which may also be related to the spatial landscape as is now been referred to.

More importantly, unlike what is stated line 154, the RFLP RT-PCR analysis do not convincingly support the RNA-seq data (in addition panels shown in Fig. S3 are barely readable): in Figure S3G the allelic bias for G6pd upon Dox treatment is limited and not much stronger than for chr12 genes in Tg-12 clones shown in S3J.

To validate the RNA-seq analysis more accurately we now quantified allele-specific gene expression by RT-PCR followed by pyrosequencing, and have implemented the new data in Figures 3 and S3.

In revised Figure S3B, *Rnf12* expression is affected by Xist induction in all tested clones and different clones show different silencing efficiency as observed in Figure 3B,D,I (e.g. clone 77 shows higher silencing efficiency compared to clones 339 and 86).

In revised Figure S3C, RT-PCR and pyrosequencing analysis confirmed that silencing of chromosome 12 genes is overall less efficient and more heterogeneous compared to X-linked silencing. In clone 55 the expression of all tested genes (*Pole2*, *Tcl1*, *Pnn* and *Glr5*) appears to be affected, confirming that clone 55 is the Tg-12 clone that shows higher degree of silencing (Figure 3E-F). Similarly, in clone 160 three genes are affected by Xist induction (*Tcl1*, *Pnn* and *Glr5*) whereas clones 228, 273 and 292 show overall poor silencing (e.g. expression levels of one or zero tested genes is affected).

Similarly, and in contrast to what the authors conclude (line 168), RFLP analysis performed on cells grown in serum+LIF doesn't reveal a clear shift in the allelic expression (Figure 3F). Strikingly, there is a strong allelic bias (~80%) for MeCP2 and G6pd (albeit to a lesser extent) toward the Cas allele in untreated cells;

We repeated the allele-specific expression analysis of X-linked genes by RT-PCR followed by pyrosequencing for three independent Tg-E clones (87, 68 and 64). We tested three X-linked genes, *Rnf12* (Figure 3F), *Abcb7* and *Pgk1* (Figure S3D) at different time point upon Xist induction. Pyrosequencing analysis is more accurate than the previously performed RFLP analysis and showed silencing of the the Cas allele for all tested genes at each time point of the analysis. We have included these findings in revised figure panels of Figures 3 and S3.

In Tg-12 clones, the Chr12 gene *Fcfl* is more expressed from the 129 allele despite the presence of 2 Cas Chr 12. Altogether, these observations raise question as to the conclusiveness of the RFLP analyses.

As suggested by this reviewer, to validate the RNA-seq data, we have performed RT-PCR followed by pyrosequencing for several chromosome 12 genes, including *Fcfl*, *Pole2*, *Tcl1*, *Nampt*, *Pnn* and *Glr5x*. The new data has now been included in Figure 3 and S3 of the revised manuscript.

2. The section "Xist-mediated inactivation of the X;8 translocation product rescues ESC clones from lethal aneuploidy" doesn't lead to an essential conclusion as such, compared to the rest of the manuscript and blocks to some extent the logical progression of the reading. The fact that inactivation of the translocated product is required for survival of cells with unbalanced translocation has been already described (although I acknowledge that the inducible system used here allows to experimentally test this hypothesis). Nevertheless, I would suggest to remove this section, while keeping the allelic analysis of Xist expression upon dox treatment and differentiation in the context of the previous section ("Xist-mediated gene inactivation efficiency is locus dependent"); this would nicely link the allelism in Xist expression to the silencing. In addition, as this previous section is already a bit long, it could be split in two with a new section starting with the neuronal differentiation of transgenic cells (from line 170).

We thank this reviewer for this suggestion and have removed most of the section describing the unbalanced translocation during ES differentiation from the manuscript and only briefly mention the Tg-X;8 results in the new section 'Xist transgene mediated silencing upon ES cell differentiation', where the effects of differentiation on gene silencing is discussed.

3. Figure 6: The link between degree of silencing and gene density around Xist insertion site is supported by the comparison of Tg-12 clones 55 and 292. However, more clones should be tested; according to this hypothesis, in Tg-12 clones 228 and 273, Xist should be integrated in a gene poor region, whereas in Tg-12 clone 160 gene density around Xist should be higher.

For the revised version of our manuscript we have now mapped most (except two) of the integration sites of our transgenes by TLA. The results of this have been implemented in Table 1 which provides an overview of all the clones analysed. Relating gene silencing efficiency to the genomic environment of the integration site, we have to conclude that on chromosome 12 variability cannot fully attributed to the gene density near the integration site, as clone 228 integrated near a gene dense area, but still shows poor gene silencing. This suggests that other factors, possibly the spatial proximity, might be instructive, and this is now mentioned in the revised discussion. Overall, comparing silencing of chromosome 12 or 8 genes, we find this to range between 15% and 26%, which is always lower than X linked genes that show silencing efficiencies in the range between 35% and 72%. This indicates that X-specific features are also in place, and in the discussion we refer to LINEs which may play a role in setting up a heterochromatic compartment that may be more efficiently compiled on LINE dense X chromosomal sequences. Also, we found a reduction of H3K27Ac at efficiently silenced genes, specifically for the X chromosome, suggesting that weaker, less active promoters are better targets for Xist. This is now discussed in the discussion of the revised manuscript.

Furthermore, the authors later refer to Tg-X clones 86 and 190 and Tg-X;8 clones 267 and 203b as showing efficient inactivation of X-linked genes. However, Tg-X clones 85 and 109 show more efficient silencing than 86 and 190 (Figure 3B); it would thus be interesting to also know their integration sites.

As mentioned above for the Tg-12 clones, we also determined the integration sites of our Tg-X clones. To our surprise in clone Tg-X 85 Xist is integrated in a telomeric region, but nevertheless shows efficient silencing, confirming that gene density in the transgene neighbourhood cannot explain long range silencing efficiency. Moreover, while adjusting the scale axis of Figure 6A,B,C according to the reviewer's suggestion, we found a script mistake that resulted in the wrong representation of clone 190 transgene along chromosome X. The right integration site is now shown in the revised version of this manuscript.

Gene density in 1 Mb bins in panels A-C should be plotted to the same scale for proper comparison.

We did as suggested.

Minor comments

Line 85: wrong reference to figure panels, should be Figure 1D-1E

Line 97-99 and Figure 2: The author should explain why they used ESC carrying a Tsix stop signal

We purposely chose to use the Tsix lox-stop-lox ES cell line as this line acquired one additional chromosome 12. This line (Luikenhuis et al. 2001) carries a floxed Tsix-stop allele that can be deleted by cre-mediated recombination. While generating our inducible system we performed cre-mediated recombination to remove the floxed selection cassette from the targeting BACs and this process also resulted in loss of the Tsix Stop cassette in all clones except clones 160 and 267. We have more specifically explained this in the results section of the revised manuscript.

Figure 2 I-K: the Xist RNA cloud appears much larger than normal, endogenous clouds. Did the authors somehow assess this?

We agree with the reviewer that Xist spreading appeared variable between clones, and indeed performed excessive confocal analysis. Unfortunately, we do not dare to conclude anything with respect to coating of the X or autosomes, as the coating was very variable even within the same clone. We therefore decided not to include these data in the manuscript.

Line 167: Figures 3H-3I: should read 3F-3G

Figure 3I, upper panel; Figure 3J, Figure 4C: description of the x axis is missing (dox/diff)

Figure 4S, 5S, 6S: should be S4, S5, S6 in the figure and in the text (line 205).

Line 207-208: "...percentage of cells that show a Xist-coated chromosome..."

We addressed and corrected all indicated mistakes in the revised manuscript.

Figure 5: Could the authors define what each category correspond to, in terms of allelic ratio?

Here we have to apologize that we cannot define the categories by a specific allelic ratio as this would make it impossible to compare X and autosomes. Categories are therefore defined by ranking.

In addition, the sentence "On chromosome 12, the distribution of silenced, partially affected and not silenced genes was invariant along the entire chromosome length regardless of Xist transgene position" (line 235-236) may be misleading, as for X-linked genes, the distribution is also similar ("invariant") in all clones. It is more that, on chromosomes 12 (and 8), silenced, partially affected and not silenced genes are evenly distributed along the chromosome in all clones, without forming blocks?

The reviewer is right and we have changed the text accordingly.

Reviewer #3 (Remarks to the Author):

Major issues:

1) Given the large number of ESC clones, readers would benefit greatly from a reference table that goes beyond Fig. 1a, and summarizes karyotype, Tg-integration coordinates (from TLA) including allele (129/Cast), and additional information (e.g.

presence of Tsix Stop allele). This supplementary table should also list the GEO sample accession number(s) for each clone.

We thank this reviewer for the positive and constructive comments, and as suggested have compiled a Table providing an overview of all the clones and clone features.

2) RFLPs after PCR are used for both clone screening and measuring allelic ratios of expression, but it's clear from the sub-optimal digestion patterns seen in Fig. 1c, S1 and S3, that PCR reactions had plateaued. Once reactions run out of primers/dNTPs, template strands are no longer copied but reanneal with reverse complement template strands. These can originate from the other allele, producing a mismatch at the polymorphic site that is resistant to restriction enzyme digestion. The over-abundance of undigested PCR product in the gel images is likely due to this effect, and compromises most of Fig. 3. Options are to repeat RFLP assay from qPCR samples (during exponential phase, or with a final extension in a fresh PCR reaction), or to perform qPCR with allele-specific primer pairs (as done for Xist in Fig. 3j and 4c).

We agree with this reviewer that reannealing of PCR products leads to overrepresentation of the undigested product. To circumvent this we use 1/10th of the final PCR reaction to perform one more extension with new dNTPs and primers. Nevertheless, to take away any concerns we redid all the allele specific PCRs by RT-PCR followed by pyrosequencing. The results of these PCRs have now been implemented in the revised manuscript and replace all RFLP based PCR results shown in the initial submission.

3) The RNA-seq data would be much more informative if discussed right away in the context of each integration site. Much of figures 3 and 4 could move into the supplement, which would allow for more extensive presentation of the RNA-seq data, and their correlation analyses (currently Figs. 5,6).

As suggested by the reviewer we have implemented the TLA analysis in the first section of the results (as part of new Table 1), and then present the RNA seq data in the context of the integration site in the third paragraph of the results section.

Superimposition of the data in Fig. 5 make it largely unreadable. Non-aggregated data from different clones (as in Fig. S6) could be presented at different distances from the integration site (1-2 Mb, 20 Mb, and chromosome-wide).

We agree with the reviewer that it is difficult to interrogate clone specific data from the panels shown in Figure 5. However, this was not intended with these plots, which was aimed to provide an overview of silencing per chromosome. In order to allow the reader to examine silencing of each individual clone we refer to revised Figure S6, which we extended by also providing data for the individual TgX;8 and Tg-12 clones.

4) Were allelic ratios calculated by summing over all genic variants or only exonic ones? Were repetitively aligning reads and transcripts excluded? The methods section refers to a previous study, but that one didn't aggregate allelic ratios across the chromosome. More detail on the bioinformatics is necessary.

The allelic counts per gene were then estimated by summing up the number of 129- and Cast- specific reads of all exonic SNPs within the gene and only unique alignments were reported for downstream analysis. As suggested by the reviewer, we added a more detailed explanation of the bioinformatic analysis in the methods section.

Plotting allelic ratios of RNA-seq against allele-specific qPCR (or redone RFLP) for a set of integration-proximal and -distal genes would be helpful.

As suggested by this reviewer above, and to provide more a reliable readout of our allele specific expression analysis, we repeated all single gene expression analyses by RT-PCR followed by pyrosequencing. The new results confirmed that our RNA-seq analysis is reliable.

5) The LINE/gene density correlations need to be checked (Fig. 6), and the figure legend is insufficient to allow interpretation.

Panels F,G,H are almost certainly incorrect (either the binning/smoothing is excessive, or a script error – compare to Fig. S7).

As suggested by this reviewer we made new figure panels displaying LINE/gene density.

The figure makes a strong point that Xist silencing can spread long-range from gene-rich regions (which by definition are LINE-poor). Conversely, gene-poor regions are enriched in LINEs and more repressed by default. The authors argue that silencing of genes in such LINE-rich domains is more efficient, but that's likely due to their already pre-repressed status.

A panel showing estimated expression (e.g. FPKM x allelic ratio) of genes from LINE-poor vs. LINE-rich regions (+/- dox) would be required to support the authors claim.

To address this reviewer's comment we assessed gene expression levels of silenced, partially affected and non affected genes in our RNA-seq datasets and we included these results in Figure S7D. Although for chromosome X we found that the efficiently silenced genes appear to be less expressed prior to Xist induction, this is not the case for autosomal genes. This shows that LINE elements can provide the necessary epigenetic environment that facilitates gene silencing independent of expression levels before Xist induction.

Minor points:

1) The presence of the Tsp5091 site on the Cast allele in the first diagram (Fig. 1a) is inadvertently misleading. This restriction site is specific to the 129 allele.

Figure 1a was changed accordingly.

2) The y-axis in the qPCR quantification data is unclear – is it directly relative to Actin B?

Indeed the quantification was done relative to B-Actin.

Where does Xist expression from its endogenous promoter fall on this scale?

As the tested ES cells were all undifferentiated, and undifferentiated ES cells do not express Xist endogenously this question is difficult to address. However, we did follow induction of Xist expression as described in the results section: ‘Overall, Xist RNA enrichment in doxycycline treated cells versus untreated cells varies from 10- to 250-fold in between different ESC lines (Figure 2A-2D). In spite of this variability, the enrichment of ectopic Xist in ES clones is either comparable or higher than the one reached by endogenous Xist upon neuronal differentiation of untreated ES cells (Figure S2A). In fact, endogenous Xist is up-regulated by 3 to 70 fold between day 2 and day 4 of differentiation, when XCI starts, compared to the Xist expression levels observed in undifferentiated ES cells, prior to XCI (Figure S2A).’ This analysis indicates that the expression observed upon dox treatment in undifferentiated ES cells is in the range of what is observed in differentiated ES cells.

Fig. S2A is unclear whether Xist induction is plotted against Actin B or endogenous Xist levels (as alluded to on pg. 3). Fixing the scale would also enable comparison across panels.

For all samples Xist expression was first normalized to Actin B and then the fold enrichment compared to day 0 was calculated, we added the fold enrichment information to Figure S2 so that the results we mentioned on page 3 are now more clear.

3) Have the authors tried to ascertain Tg copy numbers from the TLA data? Do they correlate with Xist induction?

As suggested by this and the other reviewers we have performed TLA analysis for nearly all clones. Although the exact copy number cannot be determined using TLA, an estimation can be made based on the number of integration sites, number of fusion reads and the ratio of coverage on the Tg and genome integration site. For all the lines used for this manuscript, the transgene copy number is estimated to be one copy only. An overview of the clone characteristics is provided in the new Table 1.

4) Figure legends are insufficient throughout, including Fig. 4a,b. Also, why is the skewing in non-induced Tg-X;8 clones incomplete (Fig. 4d,e), if dox-induced lethality is due to Xist silencing of the monosomic 15 Mb?

As suggested by this reviewer we carefully went through the legends and clarified the text where possible. We hope this improved the readability of the manuscript.

5) Pie graphs in Fig. 5d-f should also list gene density (or expressed gene density) across the domains A,B,C,D for comparison.

As suggested by this reviewer gene density has been implemented in this figure.

6) What about trans-effects? Are there differentially expressed genes on non-Tg bearing chromosomes in dox? The referenced trisomy 21 paper (human XIST Tg) showed significant off-target effects.

Our analysis focused on the expression levels of X-linked, chromosome 8 and chromosome 12 genes before and after Xist induction. However, to assess any possible trans-effect we plotted the distribution of the allele-specific ratios for all chromosomes except X,8 and 12 in dox-treated and untreated conditions. We observed no difference in the distribution of allele-specific ratios upon Xist induction. A summary of this analysis for Tg-E, Tg-X, Tg-X:8 and Tg-12 clones is shown below.

7) If the authors stick with RFLP analysis, these primer pairs should be listed in the supplement. TLA primers are also missing as well.

As suggested by the reviewers we have now applied pyrosequencing for allele specific validation of the RNA-seq, the oligos used for the TLA experiments are now listed in Table S3.

Reviewers' comments:

Reviewer #1 (Remarks to the Author):

Authors have done a good job revising the manuscript, providing more detailed explanations and additional analyses. Therefore, I am in support of publication of this manuscript.

Reviewer #2 (Remarks to the Author):

In the revised version, the authors have satisfactorily taken into account my comments and have also nicely addressed that of the other Reviewers. In particular they have repeated the allelic gene expression analyses using RT-PCR followed by pyrosequencing, which is more accurate than the RFLP previously used. They have also mapped most of the integration sites, which allowed them to more accurately conclude as to the influence of the genomic environment. Table 1 provides a very useful overview of all the clones. Overall the revised manuscript has been significantly improved.

Reviewer #3 (Remarks to the Author):

The revised manuscript by Loda, et al. is markedly improved, and would be recommended for publication by this reviewer, pending an additional minor but important revision (analytical only):

1) Figure legends are still insufficient for the interpretation of several experiments throughout the manuscript. Generally, they restate the interpretation, rather than give necessary experimental detail. This was especially problematic for figures 3 (eg. N & O) and 4 (A – what is this experiment?). In figure 3O, are the sample IDs switched?

2) Figure 3 established clearly that autosomal genes are nowhere nearly as efficiently silenced as X-linked genes. Yet, in Figure 5, where genes are ranked according to the degree of silencing and arbitrarily divided into 3 equal-sized groups (silenced, intermediate, and unaffected), autosomal genes are not ranked vis-à-vis all genes (incl. X-linked), but rather considered for themselves. That's comparing apples to oranges, as the "Not silenced" X-linked group of genes undergoes just as much silencing as the "Silenced" chr12- or chr8 groups. Even the panel-specific Cast-ratio ranking legend (yellow) is identical across all panels, when the shift to the left for each gene is clearly greater on the X, compared to 8 & 12. This should be addressed by ranking all genes (X, 8 & 12) together, and then categorizing them by hard thresholds (e.g. delta CAST ratio: 0-0.1 "not silenced", 0.2-0.3 "partially affected", 0.4-0.5 "silenced"). This new categorization should be carried through to the other panels in Fig. 5.

3) The authors have addressed the low impact of local LINE density on gene expression in the text, by arguing that it is specifically long-range silencing that is enhanced in cis by linkage to LINE elements. In my opinion, the strongest argument for this interpretation would be comparing the delta CAST ratio between chr 12 and chr 8 genes, because they have similar local LINE densities, but chr8 has much higher LINE density in cis due to linkage to the X. Is this the case? It's hard to tell from figure 5. This comparison is key to supporting the authors claim re: a role for LINES in supporting Xist silencing.

4) Regarding LINE densities, the authors acknowledged lower base-level gene expression in LINE-rich regions in the text, but didn't address the following point from the previous review: "A panel showing estimated expression (e.g. FPKM x allelic ratio) of genes from LINE-poor vs. LINE-rich regions (+/- dox) would be required to support the authors claim". In addition, the manuscript several times mentions the positive correlation between LINE density and silencing, but the actual correlation (Pearson) is not listed. How positive is it?

5) Finally, the authors incorrectly interpret the dox-mediated cell death of Tg-X clones and dox-mediated rescue of Tg-X;8 clones as functional chr8-trisomy being incompatible with cell survival during neuronal differentiation. I have three concerns here: (I.) as mentioned above, panel 4A is uninterpretable in its current form. (II.) If chr. 8 genes are not that efficiently silenced (see Fig. 3 & 5), why would Xist expression rescue? (III.) In contrast, the terminal 15 Mb of X-linked material is monosomic in Tg-X and Tg-X;8 clones. This region contains over 60 genes, which are very efficiently silenced by Xist. When Xist is not induced from the 8-translocated allele but from the full-length X (in Tg- X), these monosomic genes are silenced. It's much more likely that it is the loss of these X-linked gene products that causes the massive cell death in Tg-X clones. The authors should critically evaluate their interpretation.

We are happy to learn that all three reviewers found our revised manuscript significantly improved, and that two of the reviewers support publication of the manuscript in its current state. We would also like to thank reviewer three for his/her additional comments. Our answers to these minor points are listed below and all changes are highlighted in red in the manuscript text file. We are glad to have been able to address all of his/her comments by further implementing our manuscript or by providing a clearer explanation of the rationale that supports our analysis.

Reviewers' comments:

Reviewer #1 (Remarks to the Author):

Authors have done a good job revising the manuscript, providing more detailed explanations and additional analyses. Therefore, I am in support of publication of this manuscript.

Reviewer #2 (Remarks to the Author):

In the revised version, the authors have satisfactorily taken into account my comments and have also nicely addressed that of the other Reviewers. In particular they have repeated the allelic gene expression analyses using RT-PCR followed by pyrosequencing, which is more accurate than the RFLP previously used. They have also mapped most of the integration sites, which allowed them to more accurately conclude as to the influence of the genomic environment. Table 1 provides a very useful overview of all the clones. Overall the revised manuscript has been significantly improved.

Reviewer #3 (Remarks to the Author):

The revised manuscript by Loda, et al. is markedly improved, and would be recommended for publication by this reviewer, pending an additional minor but important revision (analytical only):

1) Figure legends are still insufficient for the interpretation of several experiments throughout the manuscript. Generally, they restate the interpretation, rather than give necessary experimental detail. This was especially problematic for figures 3 (eg. N & O) and 4 (A – what is this experiment?). In figure 3O, are the sample IDs switched?

We apologise for any inconsistency in the Figure legends and would like to thank this reviewer for pointing this out. As suggested, we have modified the Figures legends throughout the manuscript and added experimental details. All changes are highlighted in red in the revised figure legends.

2) Figure 3 established clearly that autosomal genes are nowhere nearly as efficiently silenced as X-linked genes. Yet, in Figure 5, where genes are ranked according to the degree of silencing and arbitrarily divided into 3 equal-sized groups (silenced, intermediate, and unaffected), autosomal genes are not ranked vis-à-vis all genes (incl. X-linked), but rather considered for themselves. That's comparing apples to oranges, as the "Not silenced" X-linked group of genes undergoes just as much silencing as the "Silenced" chr12- or chr8

groups. Even the panel-specific Cast-ratio ranking legend (yellow) is identical across all panels, when the shift to the left for each gene is clearly greater on the X, compared to 8 & 12. This should be addressed by ranking all genes (X, 8 & 12) together, and then categorizing them by hard thresholds (e.g. delta CAST ratio: 0-0.1 “not silenced”, 0.2-0.3 “partially affected”, 0.4-0.5 “silenced”). This new categorization should be carried through to the other panels in Fig. 5.

We agree with the reviewer that defining the categories of “silenced”, “partially affected” and “not silenced” genes by hard thresholds would be an appropriate alternative to perform data analysis. However, it still constitutes an arbitrary way of categorizing genes based on their degree of silencing. More importantly, we initially tried to follow this approach but realized that in our experimental setting and for the experimental questions we aimed to answer, gene ranking would provide the most reliable outcome. As stated in line 132 of the manuscript, ectopic XCI can be induced in maximum 60-70% of the cell population. The remaining 30-40% of the cells represent a strong background for the data analysis and prevents us from using a strict cut-off. We intentionally decided to rank X-linked, chromosome 12 and chromosome 8 genes in separated datasets. Even though the higher efficiency of X-linked gene silencing compared to autosomal genes is also clearly visible in Figure 5, this was already shown and discussed in Figure 3. In Figure 5, we do not compare X-linked versus autosomal silencing. Rather, we compare the distribution of our three categories of genes along chromosomes X, 8 and 12 independently of each other, to be able to link silencing features to genomic and epigenomic features. For these reasons, we believe that ranking all X-linked and autosomal genes together would not answer our questions but rather confirm again that X-linked genes are silenced more efficiently than autosomal genes. We hope we provided sufficient information to convince the reviewer that the proposed analysis is not feasible and would not provide additional information. We have also clarified the rationale behind our choice of generating three independent datasets in line 226 of the revised manuscript.

3) The authors have addressed the low impact of local LINE density on gene expression in the text, by arguing that it is specifically long-range silencing that is enhanced in cis by linkage to LINE elements. In my opinion, the strongest argument for this interpretation would be comparing the delta CAST ratio between chr 12 and chr 8 genes, because they have similar local LINE densities, but chr8 has much higher LINE density in cis due to linkage to the X. Is this the case? It’s hard to tell from figure 5. This comparison is key to supporting the authors claim re: a role for LINES in supporting Xist silencing.

We thank the reviewer for his/her suggestion to compare the delta Cast ratio between chr.12 and chr.8 genes to support the claim that LINE elements facilitate gene silencing. This comparison is shown in Figure 3F and 3I and suggests that the efficiency of Xist-mediated silencing is similar between chromosome 8 and 12 (15-25% of the autosomal genes are affected by Xist induction). However, our conclusion that LINE elements facilitate gene silencing is supported by the data shown in Figure 6I, in which we show specific enrichment of LINE elements around the TSS of strongly silenced genes for all tested chromosomes. In line 313 we state that “Xist RNA proximity may be sufficient to induce efficient local gene silencing but for regions that are further away from the Xist transcription

locus both truncated and full-length LINEs may facilitate Xist-mediated transcriptional inactivation”.

Our conclusion here is supported by the observation that in TgX;8 clones in which Xist RNA is induced from the autosomal portion of the X;8 translocation product, X-linked genes are more efficiently inactivated compared to chr.8 genes, although the latter ones are in close proximity of the Xist transcription sites. In addition, when we addressed whether there is preferential inactivation of specific genes along chromosome 8, we found a slight tendency of strongly silenced genes to be enriched in close proximity of the X;8 translocation breakpoint, several Mb away from the Xist transgene integration sites of clones 267 and 203b (line 234). This observation is now highlighted in line 236 “this tendency most likely reflects more efficient inactivation of chromosome 8 genes in proximity of the X;8 translocation breakpoint”, and discussed in line 416. Finally, the observation that genes in 4 Mb regions around the transgene integration site are always more efficiently inactivated than regions located 40 and 200 Mb from Xist further supports the conclusion that X-linked specific elements such as LINEs are important for long-range silencing.

However, we acknowledge that our conclusion in line 313 may lead to think that LINE elements are exclusively important for long-range silencing rather than for general silencing, and we changed these lines as follows (line 321 of the revised manuscript): “Taken together, our observations suggest that both truncated and full-length LINEs may facilitate Xist-mediated transcriptional inactivation of both X-linked and autosomal genes. In particular, the X-chromosome specific enrichment of LINE elements might explain the efficient long-range silencing of X-linked genes in Tg-X;8 clones 267 and 203b in which Xist RNA starts to spread from the autosomal portion of the X;8 translocation product”.

4) Regarding LINE densities, the authors acknowledged lower base-level gene expression in LINE-rich regions in the text, but didn't address the following point from the previous review: “A panel showing estimated expression (e.g. FPKM x allelic ratio) of genes from LINE-poor vs. LINE-rich regions (+/- dox) would be required to support the authors claim”. In addition, the manuscript several times mentions the positive correlation between LINE density and silencing, but the actual correlation (Pearson) is not listed. How positive is it?

As mentioned above in our answer to point 2) the induction of Xist in only 60-70% of cells limits the power of single gene analysis, because this background leads to small changes in delta ratio, partially masking the effect of Xist induction. Thus, although we agree with the reviewer that showing estimated expression of genes from LINE-poor versus LINE-rich environment would further support our claim, we believe that, in our experimental setting, the statistical analysis performed in Figure 6I provide the most reliable readout to assess the role of LINE density in Xist's silencing efficiency. To support our claims statistically, in Figure 6I, we performed Wilcoxon rank-sum tests showing that higher LINE element densities are consistently associated with the category of “efficiently silenced” genes. We did not calculate Pearson correlation coefficients because the representation of our data in independent categories is not suitable for this analysis. To avoid confusion, we have modified parts of the text in which we previously mentioned “positive correlation”.

5) Finally, the authors incorrectly interpret the dox-mediated cell death of Tg-X clones and dox-mediated rescue of Tg-X;8 clones as functional chr8-trisomy being incompatible with

cell survival during neuronal differentiation. I have three concerns here: (I.) as mentioned above, panel 4A is uninterpretable in its current form. (II.) If chr. 8 genes are not that efficiently silenced (see Fig. 3 & 5), why would Xist expression rescue? (III.) In contrast, the terminal 15 Mb of X-linked material is monosomic in Tg-X and Tg-X;8 clones. This region contains over 60 genes, which are very efficiently silenced by Xist. When Xist is not induced from the 8-translocated allele but from the full-length X (in Tg- X), these monosomic genes are silenced. It's much more likely that it is the loss of these X-linked gene products that causes the massive cell death in Tg-X clones. The authors should critically evaluate their interpretation.

We thank the reviewer for this alternative explanation of the phenotype and we have modified our interpretation of the results in line 210 as follows: "Since gain of extra chromosomes is usually better tolerated than chromosomal loss⁷⁷, silencing of the monosomic X-linked genes on the 129/Sv chromosome rather than partial trisomy of chromosome 8 is most likely responsible for the lethality observed upon differentiation of Tg-X clones." We also added details to the legend of Figure 4A.

REVIEWERS' COMMENTS:

Reviewer #3 (Remarks to the Author):

My concerns have been addressed satisfactorily and I think that in its current form the paper is acceptable.